# Pentoxifylline Enhances the Effects of Doxorubicin and Bleomycin on Apoptosis, Caspase Activity, and Cell Cycle While Reducing Proliferation and Senescence in Hodgkin’s Disease Cell Line

**DOI:** 10.3390/cimb47080593

**Published:** 2025-07-28

**Authors:** Jesús A. Gutiérrez-Ortiz, Oscar Gonzalez-Ramella, Fabiola Solorzano-Ibarra, Alejandro Bravo-Cuellar, Georgina Hernández-Flores, José A. Padilla-Ortega, Fernanda Pelayo-Rubio, Jorge R Vazquez-Urrutia, Pablo C. Ortiz-Lazareno

**Affiliations:** 1Doctoral Program in Biomedical Sciences, Centro Universitario de Ciencias de la Salud (CUCS), Universidad de Guadalajara (UDG), Guadalajara 44340, JA, Mexico; alejandro.gutierrez2329@alumnos.udg.mx; 2División de Inmunología, Centro de Investigación Biomédica de Occidente (CIBO), Instituto Mexicano del Seguro Social (IMSS), Sierra Mojada 800, Guadalajara 44340, JA, Mexico; alejandro.bravoc@imss.gob.mx (A.B.-C.); georgina.hernandezf@imss.gob.mx (G.H.-F.); 3Servicio de Hematología y Oncología Pediátrica, Hospital Civil de Guadalajara “Dr. Juan I. Menchaca”, Guadalajara 44340, JA, Mexico; oscar.gonzalez@onkokid.com; 4Departamento de Reproducción Humana, Crecimiento y Desarrollo Infantil, Centro Universitario de Ciencias de la Salud (CUCS), Universidad de Guadalajara (UDG), Guadalajara 44340, JA, Mexico; 5Instituto de Investigación en Enfermedades Crónico Degenerativas, Departamento de Biología Molecular y Genómica, Centro Universitario de Ciencias de la Salud (CUCS), Universidad de Guadalajara (UDG), Guadalajara 44340, JA, Mexico; fabiolasolorzanoibarra@gmail.com; 6Servicio de Hematología, Hospital Civil de Guadalajara “Fray Antonio Alcalde”, Guadalajara 44280, JA, Mexico; dr.padillaortega@gmail.com (J.A.P.-O.); fer.pelayo@hotmail.com (F.P.-R.); 7Department of Medicine, Penn State Hershey Medical Center, Hershey, PA 17033, USA; jvazquezurrutia@pennstatehealth.psu.edu

**Keywords:** pentoxifylline, doxorubicin, bleomycin, Hodgkin lymphoma, apoptosis, senescence, proliferation

## Abstract

Hodgkin lymphoma (HL) is a common neoplasm in adolescents and young adults, primarily treated with doxorubicin (DOX) and bleomycin (BLM), which may cause severe adverse effects. The cure rate decreases to 75% in advanced-stage disease, highlighting the need for improved treatment strategies. Pentoxifylline (PTX), an NF-κB pathway inhibitor, enhances chemotherapy-induced apoptosis in cancer cells, making it a promising candidate for HL therapy. This study assessed the effects of PTX, DOX, and BLM on apoptosis, proliferation, and senescence in Hs-445 HL cells. Cell viability and clonogenicity were measured by spectrophotometry and spectrofluorimetry, while apoptosis, caspase activity, cell cycle, mitochondrial membrane potential (ΔΨm), proliferation, and senescence were analyzed via flow cytometry. Gene expression was assessed by qPCR. PTX significantly induced apoptosis, especially when combined with BLM or BLM+DOX (triple therapy), and modulated gene expression by upregulating proapoptotic and downregulating antiapoptotic markers. PTX increased caspase-3, -8, and -9 activity and disrupted the ΔΨm, particularly with BLM or triple therapy. Furthermore, PTX abolished DOX-induced G2 cell cycle arrest, reduced proliferation, and clonogenicity, and reversed DOX- and BLM-induced senescence. In conclusion, PTX induces apoptosis in HL cells, enhances DOX and BLM cytotoxicity synergistically, and reverses senescence, suggesting its potential as an adjunct therapy for HL.

## 1. Introduction

Hodgkin lymphoma (HL) is a neoplasm derived from germinal center B lymphocytes that primarily affects the lymph nodes. It accounts for approximately 10% of all diagnosed lymphomas [1,2]. HL is characterized by Reed-Sternberg cells, which exhibit constitutive activation of the NF-κB and JAK-STAT signaling pathways. These pathways induce the expression of proinflammatory cytokines and antiapoptotic factors, which support cell survival [3,4].

Chemotherapy remains the primary treatment for HL, often involving combinations of drugs such as anthracyclines like adriamycin (ADR) and doxorubicin (DOX), as well as bleomycin (BLM) [5]. These agents cause DNA damage in cancer cells, which initiates apoptosis [6]. This process depends on a balance between the expression of proapoptotic and antiapoptotic genes and proteins belonging to the BCL-2 family. Apoptosis occurs in response to extra- or intracellular stimuli that activate death receptors and that promote mitochondrial membrane permeabilization. These events ultimately trigger caspase activation, which degrades cellular components [7].

Unfortunately, these therapeutic approaches can produce significant side effects, such as cardiotoxicity and pulmonary fibrosis, which limit treatment options and impair patients’ quality of life [8,9]. Moreover, while the cure rates for early-diagnosed patients are pretty favorable, up to 70% of patients are diagnosed at advanced disease stages, in which the response rates drop to 75% [10].

Thus, there is an urgent need to establish new therapeutic options that increase the antitumor action of chemotherapy and reduce the patients’ side effects [11,12]. In this context, drug repurposing is presented as an attractive process that identifies new applications for existing drugs, offering advantages over the development of new drugs in terms of the pharmacological advance. Repurposing existing drugs offers the benefit of already knowing their pharmacokinetic and safety profiles, significantly reducing the costs and time required for clinical implementation [13,14].

Various drugs have been identified in this context for potential repositioning in cancer treatment. For instance, pentoxifylline (PTX) [1-(5-oxyohexyl)-3,7-dimethylxanthine], a semi-synthetic derivative of methylxanthine, is a nonspecific inhibitor of phosphodiesterase and Tumor Necrosis Factor-alpha (TNF-α) with notable anti-inflammatory properties [14,15,16,17].

PTX has been shown to abrogate the phosphorylation of serines 32 and 36 of IκB, thereby inactivating the NF-κB transcription pathway. This process suppresses the expression of antiapoptotic genes that protect against cell death [18,19]. Additionally, PTX sensitizes tumor cells to ionizing radiation and enhances the cytotoxic effects of various chemotherapeutic agents [20,21,22]. For example, PTX significantly increases the tumor apoptosis induced by chemotherapeutic agents such as ADR and cisplatin in cervical cancer cells [18,23].

The efficacy of PTX is not limited to a specific tumor type. Studies have reported similar effects in cancer cell lines derived from prostate, hepatocarcinoma, breast, monocytic leukemia, cutaneous T-cell lymphoma, and colon cancers [17,24,25,26,27,28]. Notably, lymphoma-bearing mice treated with PTX and ADR demonstrated survival exceeding one year, even when administered half the standard therapeutic dose of ADR [17].

This study aimed to evaluate the effects of PTX, DOX, BLM, and their combinations on apoptosis, caspase activity, cell cycle, cell senescence, cell proliferation, and mitochondrial membrane potential in HL cells.

## 2. Results

### 2.1. Determination of the IC_50_ of PTX, DOX, and BLM in the Hs-445 HL Cell Line

To calculate the IC_50_, the percentage of cell viability for each dose–response curve was measured employing the control group (an untreated baseline group), which represented 100% viability. As depicted in Figure 1a, cell viability decreased dose-dependently with PTX at 24 and 48 h. Viability reached 53% ± 4.9 and 45.2% ± 2.4 with 4 mM and 8 mM doses, respectively. Therefore, 8 mM was selected as the final IC_50_.

DOX treatment produced a pronounced effect after 48 h of exposure, reducing cell viability in a dose-dependent manner. At a concentration of 1 µM, viability decreased to 45.4% ± 3.5, making this dose the final IC_50_ (Figure 1b).

According to the cell viability data obtained using the WST-1 assay, the IC_50_ values were 8 mM for PTX and 1 µM for DOX. In the case of BLM, although the 100 and 200 mU doses resulted in a more pronounced reduction in viability after 48 h (64.5% ± 0.8 and 61.2% ± 2.4, respectively), an IC_50_ was not reached under these experimental conditions (Figure 1c).

Nevertheless, based on the literature, the 100 mU concentration of BLM was selected for further evaluation by flow cytometry using the same methodology applied in the apoptosis assays to determine the percentage of cell death. In this analysis, it was observed that this concentration induced cell death close to 50%, and therefore, this concentration was designated as the IC_50_ for use in subsequent assays.

Regarding the effect of the pharmacological interaction of the drugs tested, Figure 2a displays a 3D representation of the dose–response of the percentage of cell viability inhibition achieved with the combined treatments at different concentrations. The observed inhibition rates were 57.74% for PTX + DOX, 67.73% for PTX + BLM, 57.49% for DOX + BLM, and 70.47% for the triple combination PTX + DOX + BLM, using the concentrations established in the study (PTX 8 mM, DOX 1 µM, and BLM 100 mU) (Appendix A).

Figure 2b presents the synergy analysis based on the HSA model for the combinations of DOX (0, 1, 2, 4, and 6 µM) and BLM (0, 25, 50, 100, and 200 mU) with PTX at a fixed concentration of 8 mM. The mean synergy score was 13.434, indicating that the combination of PTX (8 mM) with DOX (1, 2, 4, and 6 µM) and BLM (25, 50, 100, and 200 mU) results in an overall synergistic effect.

Figure 2c summarizes the mean synergy scores obtained for the different drug combinations tested. The PTX + DOX combination yielded a mean score of 6.2 (across all tested concentrations of PTX and DOX), the DOX + BLM combination obtained −3.4 (across all tested concentrations of DOX and BLM), and the PTX + DOX + BLM combination resulted in 9.7 (across all tested concentrations of PTX, DOX and BLM). According to the HSA model criteria, all three combinations fall within the range of additive interaction. In contrast, the PTX + BLM combination achieved a mean score of 11.8 (across all tested concentrations of PTX and BLM), indicating a synergistic interaction.

Specifically, for the experimental concentrations used in this study (PTX 8 mM, DOX 1 µM, and BLM 100 mU), an additive effect was observed for PTX + DOX (−0.26) and DOX + BLM (1.39), and a synergistic effect for PTX + BLM (13.91) and the triple combination PTX + DOX + BLM (17.79). All synergy score values for the pharmacological analysis of the combinations evaluated are provided in (Appendix A).

Collectively, these results indicate that the combination of the three drugs exerts a predominantly synergistic effect, enhancing the overall cytotoxic response.

### 2.2. PTX Induces Apoptosis in Hs-445 Cells and Enhances the Apoptosis Induced by DOX and BLM

The effects of the treatments on the induction of apoptosis in Hs-445 cells after 48 h were evaluated. Appendix A and Figure 3a depict the gating strategy followed in this assay and the representative density plots distinguishing live (L), early-apoptotic (EA), late-apoptotic (LA), and necrotic cells (N) based on Yo-Pro and Zombie Aqua staining.

As illustrated in Figure 3b, PTX (43.3% ± 2.8), DOX (56.4% ± 1.3), and BLM (58.8% ± 2.4) individually reduced the percentage of live cells significantly compared to the untreated baseline group (84.6% ± 2.1; *p* < 0.05). Surprisingly, PTX alone led to a significant reduction in live cells compared to DOX (*p* < 0.05) and BLM (*p* < 0.05). The group exposed to the combination of PTX + BLM (31.6% ± 3.5) and the triple combination of PTX + DOX + BLM (31.1% ± 2.1) achieved the lowest percentage of live cells compared to those treated with DOX + BLM (45.4% ± 2) and their respective individual treatments (DOX or BLM alone) (*p* < 0.05).

A significant increase in the early-apoptotic cells was observed in the PTX group (27.7% ± 5.8; *p* < 0.05) and in the PTX + DOX + BLM group (31.1% ± 1.1; *p* < 0.05) compared to the baseline group (11% ± 2.2). Furthermore, cells treated with PTX + BLM (31.2% ± 4.8) showed a significant increase in early apoptosis compared to BLM alone (22.7% ± 4.2; *p* < 0.05). Similarly, the PTX + DOX + BLM combination exhibited a notorious difference compared to DOX + BLM (18.6% ± 1.3; *p* < 0.05).

Late-apoptotic cell percentages were noticeably higher in all groups with regards to the baseline group (3.1% ± 0.1; *p* < 0.05), except for the PTX + DOX group. Additionally, a distinction was noted between the BLM group (13.9% ± 1.9) and the PTX + BLM group (28.9% ± 2.4; *p* < 0.05).

Although necrotic cell percentages were generally low, the DOX + BLM treatment (10.6% ± 0.5) induced significantly more necrosis than in the untreated group (1.3% ± 0.3; *p* < 0.05) and in the PTX + DOX group (2.9% ± 1.7; *p* < 0.05).

Therefore, PTX induces apoptosis in Hs-445 cells and enhances the percentage of cell death when combined with DOX and BLM. The combinations of PTX + BLM and the triple treatment of PTX + DOX + BLM produced the most pronounced apoptotic effects.

### 2.3. The Combination of PTX, DOX, and BLM Increases the Cleavage of Caspases-3, -8, and -9 in Hs-445 Cells

Caspases are crucial mediators of apoptosis. Consequently, the effects of PTX, DOX, BLM, and their combinations on the activation of caspases-3, -8, and -9 in Hs-445 cells were assessed, as depicted in the representative histograms in Figure 4a.

The individual treatments significantly enhanced caspase-3 activity compared to the untreated baseline group (11.9% ± 1.4; *p* < 0.05) (Figure 4b). The combined PTX + BLM treatment amplified this effect, achieving maximal functionality with a 5.76-fold increase compared to the untreated baseline group and by 1.65-fold increase (68.2% ± 8.2) concerning PTX alone (41.3% ± 4) (*p* < 0.05).

As shown in Figure 4b, PTX, DOX, and BLM individually increased caspase-8 and -9 activity compared to the untreated baseline group (*p* < 0.05), with BLM exhibiting the most pronounced effect. Maximal activation of caspases -8 and -9 occurred with the PTX + BLM combination and with the triple treatment of PTX + DOX + BLM, both exceeding the baseline group and individual or double treatments (*p* < 0.05). Notably, the PTX and BLM combination enhanced caspase-8 and -9 activity by 6.78- and 6.92-fold, respectively, compared to the baseline group (*p* < 0.05). Additionally, the triple treatment further increased caspase-8 (60.1% ± 6.2, 1.98-fold) and caspase-9 (57.9% ± 3.6, 1.56-fold) activity with regards to the PTX + DOX group (*p* < 0.05).

Therefore, these findings suggest that PTX enhances BLM- and DOX-induced caspase activity, with the most substantial effects occurring in the PTX + BLM and PTX + DOX + BLM groups.

### 2.4. PTX and BLM Alone or Combined Induce Mitochondrial Depolarization in Hs-445 HL Cells

The mitochondria play a vital role in the apoptosis process. To assess whether PTX, DOX, BLM, or their combinations caused a loss of ΔΨm in Hs-445 cells, JC-10 dye was used. In healthy cells, the dye accumulates in the mitochondrial matrix as J-aggregates, emitting red fluorescence. However, JC-10 diffused out of the organelle and shifted to a monomeric form in cells with compromised mitochondrial membrane integrity, exhibiting green fluorescence. This process is illustrated in the gating strategy and the representative density plots in Appendix A and Figure 5a.

As observed in Figure 5b, PTX alone and all other treatment groups caused a substantial shift toward green fluorescence compared to the untreated baseline group (*p* < 0.05), indicating mitochondrial depolarization. Among the individual treatments, PTX (54.8% ± 0.7) and BLM (56.5% ± 1.9) induced a more pronounced depolarization concerning the DOX group (44.1% ± 5.4) or the baseline group (22.8% ± 1.9) (*p* < 0.05).

The PTX + DOX combination gave rise to a significant loss in ΔΨm (53.3% ± 0.4) compared to DOX alone (*p* < 0.05). Similarly, the PTX + BLM treatment led to a more pronounced depolarization (67.7% ± 2.7) than PTX or BLM alone (*p* < 0.05). Furthermore, cells subjected to the triple treatment with PTX + DOX + BLM resulted in the highest depolarization (70.1% ± 2.4), exceeding the PTX + DOX group (*p* < 0.05) and the DOX + BLM group (59.9% ± 3.7) (*p* < 0.05).

These results implicate the involvement of the mitochondrial pathway in PTX- and BLM-induced cell death, as evidenced by significant changes in ΔΨm in the Hs-445 cells.

### 2.5. PTX Abrogates Cell Arrest in the G2 Phase Induced by DOX

To elucidate whether the treatments modulated the cell cycle, Hs-445 cells were treated with PTX, DOX, BLM, and their combinations for 48 h. Figure 6a presents representative histograms illustrating cell distribution across the G1, S, and G2 phases.

In the untreated baseline group, 73.3% ± 1.8 of cells were in the G1 phase (Figure 6b), a percentage similar to that observed with BLM (73.9% ± 2), with no significant difference between these groups. Furthermore, PTX treatment increased the proportion of cells in the G1 phase to 79.5% ± 0.5, while the group exposed to DOX decreased it to 59.8% ± 2.3, significantly lower than in all other groups (*p* < 0.05). Moreover, the triple treatment (PTX + DOX + BLM) elevated the G1-phase percentage to 77.9% ± 1.5, significantly higher than the baseline group (*p* < 0.05). No significant differences were detected in the S phase.

Concerning the G2 phase, DOX alone markedly increased cell arrest to 35.5% ± 2.6, compared to all the treatments (*p* < 0.05). PTX reduced this proportion to 16.3% ± 0.6. Indeed, the PTX + DOX combination further decreased G2 arrest to 17.6% ± 0.6, significantly lower than employing DOX and PTX alone (*p* < 0.05). The lowest G2 phase percentage, 16.2% ± 2.5, was achieved with the triple treatment of PTX + DOX + BLM, which was significantly lower compared to DOX (*p* < 0.05) and BLM alone (*p* < 0.05).

Hence, these findings indicate that DOX alone significantly induces G2-phase arrest. At the same time, PTX, alone or in combination, reduced the G2 phase percentage and increased the proportion of cells in the G1 phase, thereby abrogating the G2 arrest induced by DOX.

### 2.6. PTX Exhibits Antiproliferative Activity in Hs-445 Cells

One of cancer cells’ most essential characteristics lies in their ability to maintain persistent proliferation. Therefore, to evaluate the antiproliferative effects of PTX, DOX, BLM, and their combinations, Hs-445 cells were labeled with the thymidine analog EdU, which binds to a fluorescent azide for detection. The gating strategy followed in this assay and the representative density plots of the experimental groups are displayed in Appendix A and Figure 7a.

As expected, the baseline group, without treatment, revealed the highest proliferation rate at 28.5% ± 2.4 compared to all other treatments (*p* < 0.05) (Figure 7b). Additionally, PTX (1.7% ± 0.3), DOX (8.6% ± 0.9), and BLM (1.8% ± 0.2) individually reduced cell proliferation, with PTX and BLM exhibiting a more pronounced reduction compared to the DOX-treated group (*p* < 0.05).

The PTX + DOX combination also reduced proliferation to 2.1% ± 0.7, significantly lower than with DOX alone (*p* < 0.05). Similarly, PTX + BLM achieved an even more substantial effect, decreasing cell proliferation to 0.5% ± 0.2, notoriously lower regarding the groups treated with PTX or BLM alone (*p* < 0.05). Likewise, the PTX + BLM treatment and the triple combination of PTX + DOX + BLM (0.6% ± 0.2) exhibited the highest antiproliferative activity compared to the PTX + DOX and DOX + BLM groups (*p* < 0.05).

Consequently, the results indicate that PTX significantly inhibits tumor proliferation, demonstrating an efficacy comparable to BLM and exceeding that of DOX. The most pronounced antiproliferative effects were achieved with the PTX + BLM combination and triple therapy.

### 2.7. PTX Decreases DOX- and BLM-Induced Senescence in Hs-445 Cells

Senescence was evaluated by measuring β-galactosidase in HL cells treated for 48 h with PTX, DOX, BLM, and their combinations (Figure 8a). As depicted in Figure 8b, PTX treatment resulted in minimal senescence (2.3% ± 1.1), a value six times lower than that observed in the untreated baseline group (13.9% ± 0.4; *p* < 0.05).

As expected, the number of β-galactosidase-positive cells increased significantly in groups treated individually with DOX (21.2% ± 5.2; *p* < 0.05) and BLM (11% ± 2.7; *p* < 0.05) compared to PTX, representing 9.1- and 4.7-fold increases, respectively.

The concomitant use of PTX with DOX (2.5% ± 1.4) or PTX + BLM (4% ± 1.8) significantly reduced senescence, corresponding to 8.6- and 2.7-fold decreases relative to their respective individual treatments (*p* < 0.05). Furthermore, the percentage of senescent cells in the PTX group was 5.2 times lower than in cells treated with the DOX + BLM combination (*p* < 0.05).

Notably, the triple treatment of PTX + DOX + BLM (4% ± 1.6) substantially reduced senescence. The number of β-galactosidase-positive cells was 5.3 times lower compared to the DOX group (*p* < 0.05), 2.7 times lower than in the BLM group (*p* < 0.05), and 3 times lower regarding the DOX + BLM group (*p* < 0.05).

These findings suggest that PTX exerts the most potent inhibitory effect on DOX- and BLM-induced senescence in Hs-445 cells.

### 2.8. The Combination of PTX + BLM Reduces Tumor Colony-Forming Capacity

The effect of PTX, DOX, BLM, and their combinations on the clonogenic capacity of HL cells was evaluated after 48 h of treatment. As shown in Figure 9, the untreated baseline group retained a high colony-forming capacity, as evidenced by elevated fluorescence emission (145.2 ± 4.8 RFU) compared to the treated groups (*p* < 0.05).

All treatments significantly reduced the clonogenic capacity, with a moderate decrease observed with the single PTX treatment (115.7 ± 3.44 RFU), although with a statistically significant difference concerning the baseline group (*p* < 0.05). However, the most pronounced inhibitory effect was observed with the PTX + BLM combination (98 ± 4.29 RFU), which significantly reduced colony formation relative to the baseline group (*p* < 0.05), as well as in comparison to the PTX + DOX (111 ± 4.64 RFU; *p* < 0.05), DOX + BLM (107 ± 4.14 RFU; *p* < 0.05), and even compared to the triple combination PTX + DOX + BLM (103.8 ± 2.5 RFU; *p* < 0.05).

These results suggest that PTX alone significantly affects the clonogenic capacity of HL cells and that this effect is enhanced when combined with chemotherapeutic agents, particularly BLM.

### 2.9. PTX Treatment Upregulates Proapoptotic Gene Expression

The effects of PTX, DOX, and BLM on the expression of genes related to apoptosis, senescence, and signaling pathways in HL cells were evaluated by qPCR.

As shown in Figure 10, all treatments predominantly induced a significant increase in the expression of proapoptotic genes, particularly *BAX* and *CASP8*, with Log2FC values ranging from 1.26 to 5.9 compared to the control baseline group (*p* < 0.05), with the most pronounced effect observed in the PTX-treated group. *BAK1* expression also increased significantly in the groups treated with PTX, PTX + DOX, PTX + BLM, and the triple combination of PTX + DOX + BLM, reaching a maximum Log2FC of 3.28 with PTX alone (*p* < 0.05). Similarly, *CASP9* was significantly overexpressed, especially in the PTX, PTX + BLM, and triple combination groups, evidenced by Log2FC values up to 3.78 (*p* < 0.05).

For *CASP3*, significant overexpression was observed only in the PTX, BLM, and PTX + BLM groups, with the highest Log2FC (3.07) in the PTX + BLM group (*p* < 0.05) compared to the baseline group. In contrast, *TP53* expression was significantly downregulated in the PTX, DOX, and triple combination groups, with a Log2FC as low as −5.04 (*p* < 0.05).

Regarding the antiapoptotic genes, the control baseline group exhibited a significant overexpression of *BIRC5/SURVIVIN* (Log2FC of 2.34; *p* < 0.05), which was significantly downregulated by all treatments, particularly with the triple combination (Log2FC of −4.6; *p* < 0.05). Likewise, *BCL2* was overexpressed in the baseline, BLM, and PTX + DOX groups, but its expression was significantly reduced in the PTX, PTX + BLM, and triple treatment groups, with the lowest expression observed in the latter (Log2FC: −4.04; *p* < 0.05). *BCL2L1/BCL-XL* expression decreased significantly in all treatment groups except in DOX and PTX + DOX, with the lowest Log2FC recorded in the PTX group (−8.8; *p* < 0.05).

Conversely, *CDKN1A/P21* expression increased significantly only in the DOX-treated group (Log2FC: 2.32; *p* < 0.05) compared to the other experimental groups but was markedly reduced in the remaining treatment groups, especially with PTX, PTX + BLM, and the triple combination, with Log2FC values as low as −4.68 compared to both the baseline and DOX groups (*p* < 0.05). No significant changes were observed in the expression of *CDKN2A/P16*, *BAD*, *JUN*, *AKT1*, or *RELA/P65*.

Overall, PTX, PTX + BLM, and PTX + DOX + BLM treatments upregulated proapoptotic genes and downregulated antiapoptotic genes, suggesting a predominant proapoptotic effect.

## 3. Discussion

Since its approval for clinical use in 1985 in the United States, PTX has been extensively studied for its properties and medical applications beyond microcirculation-related conditions [29]. Notably, its antitumor capacity has drawn significant interest, with numerous studies demonstrating its ability to potentiate the cytotoxic effect of various chemotherapeutic agents and radiotherapy. This phenomenon is attributed to this drug’s impact on multiple signaling pathways, including the downregulation of STAT3, the modulation of the MAPK and ERK 1/2 pathways, and the inhibition of the NF-kB pathway [19,30,31].

Clinical oncology has focused on advancing and utilizing chemotherapeutic drugs that efficiently eliminate tumor cells through apoptosis [32]. Therefore, the results of this study demonstrate that PTX exhibits antitumor activity, enhances the cytotoxicity of BLM and DOX, and inhibits the senescence induced by DOX in Hs-445 cells.

PTX alone emerges as a potent inducer of apoptosis, surpassing the activity of DOX and BLM when employed individually. Additionally, PTX enhanced the cell death induced by these chemotherapeutic agents, especially in combination with BLM and in the triple treatment regimen (PTX + DOX + BLM).

PTX’s antitumor activity has been observed across various tumor cell lines, including hepatocellular carcinoma, breast cancer, and human melanoma [30,33,34]. Furthermore, a synergistic effect on tumor apoptosis has been demonstrated when PTX is used in combination with other drugs such as simvastatin, doxorubicin, docetaxel, and carboplatin, surpassing the efficacy of single drug treatments [35,36,37,38] (Appendix A). These findings are consistent with our results, which showed a predominantly synergistic interaction when PTX was combined with DOX and BLM.

The mechanism by which PTX induces apoptosis involves the regulation of apoptosis-associated genes. PTX upregulates proapoptotic genes such as *DIABLO*, *DR4*, *BAD*, *NOXA*, *TP53*, *PUMA*, and *TRAIL* and increases *FAS* expression at mRNA and surface levels. Simultaneously, it downregulates the expression of antiapoptotic genes, including c-*FLIP*, *c-IAP*, and *MCL-1*, as well as the protein and mRNA levels of *BCL-XL* and *BCL-2L* [18,23,25,27]. These results align with our data, which showed a significant decrease in the expression of antiapoptotic genes such as *BCL2*, *BCL2L1/BCL-XL*, and *BIRC5/SURVIVIN*.

Meza-Arrollo et al. reported that PTX increased the expression of genes associated with the extrinsic apoptosis pathway in pediatric leukemia cells, such as *TNFRSF10*, *TNFRSF25*, *FADD*, and *CASP10*. This pattern generally, though not exclusively, favors increased proapoptotic gene expression and reduced antiapoptotic gene expression [39]. Consequently, PTX-induced apoptosis likely results from a balance in pro- and antiapoptotic gene expression, with a bias toward cell death. Therefore, a similar mechanism of action is proposed for PTX-induced apoptosis in our study model.

Although PTX has been shown to sensitize tumor cells to the cytotoxic activity of various chemotherapeutic agents, as far as we are aware, the use of PTX in HL cells or its combination with BLM (PTX + BLM) within the context of cancer treatment has not been previously explored. Our findings highlight the high efficiency of this combination, nearly matching the levels observed with the triple therapy with PTX + DOX + BLM, demonstrating that the concomitant use of PTX and BLM induces a significant synergistic effect that enhances their cytotoxicity. Therefore, to the best of our knowledge, our data demonstrate, for the first time, PTX’s potential as both an inducer and an enhancer of apoptosis in this neoplasm.

Caspases, a group of conserved enzymes within the cysteine proteases family, are pivotal in regulating and executing apoptosis [7]. Consistent with this, our results confirm the role of caspases in tumor apoptosis, revealing increased activity when cells are treated with PTX alone. Furthermore, combining PTX with DOX and BLM enhances the activity of caspase-8 and -9. Notably, the combination of PTX + BLM achieves activity levels comparable to those observed with the triple treatment.

These results align with those of previous research demonstrating that PTX boosts the activity of caspases-3, -8, and -9 [19,33] and amplifies chemotherapy-induced activation [18,23,26]. Additionally, PTX also increases the gene expression of these enzymes [38,39], a finding that was also observed in our model for caspases-8 and -9.

In our analysis of caspase-3 activity, we observed an increase with the addition of DOX to various treatment groups, though the values did not reach the magnitudes reported in previous studies [23]. Nonetheless, antitumor activity remained consistent across all treatments. This phenomenon suggests that other effector caspases, such as caspase-6 or -7, may also contribute to apoptosis in our study model [40]. Indeed, prior investigations have documented significant activation of caspase-6 and -7 in breast and prostate cancer-cell lines treated with DOX [41,42,43].

In turn, mitochondria are critical in maintaining cell integrity and viability. During early apoptosis, the permeabilization of the outer mitochondrial membrane leads to the loss in the ΔΨm, which represents a “point of no return” that results in inevitable cell death [44].

Studies have reported that PTX induces a dose-dependent loss of ΔΨm, which is further amplified when combined with other molecules, such as the proteasome inhibitor MG132 and perillyl alcohol [25,27,45]. Additionally, PTX upregulates the expression of the proapoptotic proteins BAX and BAK, members of the Bcl-2 family, which oligomerize and create pores in the mitochondrial membrane, triggering depolarization [18,38,46]. These observations are consistent with our results, which demonstrate that PTX induces substantial ΔΨm loss and enhances the depolarization caused by chemotherapy. The most significant effect was observed with the PTX + BLM combination and the triple treatment of PTX + DOX + BLM, potentially explained by the increased expression of *BAX* and *BAK1* genes in our model.

The results from the caspase activity and the ΔΨm assays closely mirror those observed in the apoptosis evaluation, demonstrating consistency across these analyses. Based on these findings, we propose that PTX induces cell death primarily through apoptosis involving both the extrinsic and the intrinsic pathways. This event is supported by the activation of the initiator caspases (caspases-8 and -9) and the observed disruption of ΔΨm. These events ultimately culminate in the cleavage of effector caspase-3 or alternative pathways, such as caspase-6 or -7, leading to apoptotic cell death. This mechanism is further supported by the PTX-induced gene expression profile, which includes the upregulation of proapoptotic genes associated with the intrinsic pathway (*CASP9*, *BAK1*, and *BAX*), the extrinsic pathway (*CASP8*), and the effector caspase (*CASP3*), as well as the downregulation of antiapoptotic genes such as *BCL2*, *BCL2L1/BCL-XL,* and *BIRC5/SURVIVIN*. Although a reduction in the expression of other proapoptotic genes, such as TP53, was observed, PTX treatment generally promotes a transcriptional profile that favors apoptosis in tumor cells.

One of the main hallmarks of cancer is the dysregulation of the cell cycle, which promotes sustained proliferative signaling and uncontrolled tumor growth [47]. Consequently, developing and using drugs targeting the cell cycle machinery has emerged as a critical therapeutic strategy in oncology [48].

Our results indicate that PTX counteracts the G2 phase arrest induced by DOX, favoring the accumulation of cells in the G1 phase across all PTX-containing treatments. These results are similar to those previously reported, revealing that PTX increases the proportion of cells in the G1 phase dose-dependently [28,33,34,37]. In contrast, DOX induces a G2 phase arrest, which is reversed when PTX is added to the treatment regimen [36,49].

PTX is known to induce G1 phase arrest by downregulating cyclin D1 and CDK6 expression levels [30,50]. Additionally, it modulates the levels of cyclin B1 and p34cdc2 [51,52]. The cyclin D1/CDK6 complex is essential for the G1-to-S phase transition, while cyclin B1 and p34 cdc2 are key components of the maturation-promoting factor that controls the G2/M transition [53].

Our findings suggest that PTX has a similar mechanism of action in Hs-445 cells, influencing the cell cycle in a manner consistent with its cytotoxic effects on tumor cells. It has been postulated that PTX, through the inhibition of the G2 phase checkpoint, reduces the time available for DNA damage repair caused by chemotherapy. As a result, cells may enter mitosis prematurely and inappropriately, potentially leading to mitotic catastrophe and subsequent apoptosis [54]. Additionally, drugs that disrupt G2 phase arrest have been demonstrated to enhance the effectiveness of specific chemotherapeutic agents and ionizing radiation by increasing tumor cell sensitivity to these treatments [55].

By inducing apoptosis in tumor cells and altering the cell cycle, PTX directly impacts cell proliferation. This antiproliferative activity has been demonstrated in melanoma and hepatocellular carcinoma cells [33,56], aligning with our findings and demonstrating that PTX’s antiproliferative capacity is comparable to that of BLM and surpasses the effects of DOX.

The clonogenicity of a cell refers to its ability to form colonies. In the context of tumor cells, clonogenicity assays are used to evaluate this capacity and to estimate toxicity and sensitivity to chemotherapeutic agents [57]. PTX alone has been shown to moderately reduce clonogenic capacity; however, it significantly potentiates the inhibition of colony formation when combined with other antitumor agents. This background is consistent with our findings, which evidence a marked decrease in clonogenicity with the combination of PTX and BLM. This effect may translate into a reduced population of tumor cells with treatment-resistant properties and high long-term proliferative potential that may contribute to disease progression [58,59].

Senescence, a stable cell cycle arrest, was traditionally viewed as a physiological tumor-suppressive mechanism due to its role in limiting cancer cell proliferation [60]. Several chemotherapeutic drugs, including anthracyclines, cisplatin, and bleomycin, are known to induce senescence [61].

Nevertheless, recent findings suggest that senescent cells may paradoxically contribute to tumor progression and relapse by promoting the proliferation of premalignant and malignant cells and enhancing tumor vascularization [60]. This effect is attributed to the release of various factors, including cytokines, growth factors, chemokines, and metalloproteases, which reshape the tumor microenvironment into a protumor state [62].

In this study, we show that PTX does not induce senescence in HL cells and that it inhibits senescence triggered by DOX. These results are in alignment with reports demonstrating PTX’s capacity to counteract the senescence induced by anthracycline, cisplatin, and docetaxel in cervical- and prostate-cancer cells. PTX also reduces the expression of p16, a protein strongly associated with senescence, promoting long-lasting growth arrest [18,23,37,62]. In our study model, no significant changes in *CDKN2A/p16* gene expression were observed; however, a significant reduction in *CDKN1A/p21*, another key marker associated with the senescent phenotype, was evident [62]. This finding is particularly relevant given that DOX treatment markedly increased *CDKN1A/p21* expression, consistent with its pro-senescent effect.

Interestingly, PTX treatment also reduced the expression of *TP53*, a tumor suppressor gene with proapoptotic functions that also regulate cellular senescence [63,64]. This downregulation may contribute to the inhibition of the senescent phenotype in HL cells beyond its direct involvement in apoptosis, especially considering that this effect is accompanied by the upregulation of other proapoptotic genes. These observations suggest a potential mechanism by which PTX may reverse senescence; however, further studies are needed to elucidate the exact pathways involved in this process.

These findings are significant because antitumor therapies that induce apoptosis rather than senescence are preferred in the treatment of cancer. Apoptosis effectively eliminates tumor cells, reducing the risk of disease progression associated with a senescent-cell phenotype. PTX stands out as a therapeutic agent capable of achieving this therapeutic advantage. Indeed, Reed-Sternberg cells exhibited a senescence-associated phenotype that contributes to disease pathogenesis by promoting a proinflammatory microenvironment through cytokine secretion, which recruits various cell types that enable disease progression [65,66]. Therefore, the reversal and subsequent elimination of these senescent tumor cells may comprise a therapeutic mechanism for treating HL. This approach may improve treatment outcomes and enhance the patient’s prognosis by directly targeting cells potentially involved in lymphoma progression.

PTX, beyond its antitumor effects and ability to enhance the cytotoxicity of chemotherapeutic agents in malignant cells, has been reported to mitigate adverse effects, particularly those associated with DOX and BLM.

In murine models, PTX has demonstrated cardioprotective properties by reversing DOX-induced myocardial damage, as evidenced by the reduced expression of inflammatory and apoptotic markers in cardiomyocytes [67,68]. In mice with BLM-induced pulmonary fibrosis, PTX treatment significantly improved alveolar lesions, decreased collagen deposition, and reduced fibrosis indices. It also downregulated the expression of genes associated with fibrosis, underscoring its potential as an antifibrotic agent with promising clinical applications [69]. Furthermore, it has been shown that the use of PTX, even at high concentrations, is not cytotoxic to normal lymphocytes from healthy donors [27].

Furthermore, Meirovitz et al. [70] observed, in a clinical trial, that patients with metastatic colon cancer receiving chemotherapy combined with PTX experienced a decrease in the frequency of stomatitis, a common chemotherapy-related side effect. These patients also exhibited improved overall survival and significantly reduced serum levels of inflammatory cytokines and tumor markers, such as the carcinoembryonic antigen, compared to the control group without PTX [70,71].

Although this study provides strong evidence of the effects of PTX, DOX, and BLM on the Hs-445 cell line, it is important to acknowledge its limitations. In particular, using a single cell line, though an appropriate model for the study of HL, may limit the generalizability of the results. Therefore, further studies in other cell models, such as KM-H2 or L-428 cell lines, are needed to provide a more robust insight into the observed pharmacological effects.

Future in vivo studies are important to examine the effects of the PTX, DOX, and BLM combination in a more complex physiological context, such as preclinical studies in murine models, to evaluate survival outcomes and tumor size reduction. Additionally, clinical trials involving patients with HL treated with PTX alongside chemotherapy are essential to assess the impact of this combination on clinical response, survival, and treatment-associated side effects. These studies would contribute to a better understanding of the therapeutic potential of these drugs in HL.

## 4. Materials and Methods

### 4.1. Cell Line

The Hodgkin’s Disease cell line Hs-445 (ATCC HTB-146, Manassas, VA, USA) was used in this study. Cells were cultured in RPMI-1640 medium (GIBCO™, Invitrogen Co., Carlsbad, CA, USA) supplemented with 10% heat-inactivated fetal bovine serum (GIBCO) and penicillin/streptomycin (GIBCO), designated RMPI-S. Incubation was carried out at 37 °C in a humidified atmosphere containing 95% air and 5% CO_2_.

### 4.2. Drugs

PTX (Sigma-Aldrich, St. Louis, MO, USA) was dissolved in RPMI-1640 to prepare a 200 mM stock solution before being used. DOX (PiSA^®^ Farmacéutica, Coyoacán, CDMX, Mexico) and BLM (Celon Labs, Hyderabad, TG, India) were dissolved in sterile 1X PBS to obtain concentrations of 4200 µM and 15,000 mU, respectively (Table 1). All drugs were stored at 4 °C for up to 3 months.

### 4.3. Cell Culture and Experimental Conditions

Hs-445 cells were seeded in T75 flasks containing RPMI-S at 4 × 10^5^ cells/mL. Fresh medium was replaced every 2–3 days. For viability assays using a colorimetric assay, 4 × 10^4^ cells/well were plated in 96-well plates with a final volume of 200 µL and treated with PTX (1, 2, 4, 8, and 16 mM), DOX (0.5, 1, 2, 4, and 6 µM), and BLM (5, 10, 15, 20, 25, 50, 100, and 200 mU) for 24 and 48 h. The final concentrations for subsequent experiments were PTX (4 mM), DOX (1 µM), and BLM (100 mU). For the apoptosis, cell cycle, caspase, mitochondrial membrane potential (ΔΨm), senescence, clonogenicity, and cell proliferation assays performed using flow cytometry, 1.5 × 10^5^ cells/well were seeded in 24-well plates with a final volume of 1 mL and treated with PTX (8 mM), DOX (1 µM), BLM (100 mU), as well as their respective combinations for 48 h. In all experiments, cells were preincubated with PTX for 1 h prior to adding DOX and/or BLM.

### 4.4. Dose–Response Curves to Determine IC_50_ and the Effect of Pharmacological Interaction

To establish the half-maximal inhibitory concentration (IC_50_), cell viability was determined using the WST-1 reagent (BioVision Research, Mountain View, CA, USA). This assay relies on the cleavage of the tetrazolium salt WST-1 into formazan by the mitochondrial dehydrogenase enzyme. Cells were treated with PTX, DOX, and BLM for 24 and 48 h under the previously described conditions. Briefly, 10 µL of WST-1 was added to each well 3 h before the end of the respective treatment. Absorbance was measured at 450 nm using a microplate reader (Synergy™ HT; Biotek, Winooski, VT, USA). Data were expressed as the percentage of cell survival relative to the control group (untreated baseline group), which was considered as 100%. Dose–response curves were generated, and the data were interpolated to identify the final concentration at which 50% cell viability was achieved.

The SynergyFinder web application was used to evaluate whether the experimental drug combinations of PTX, DOX, and BLM exhibited antagonistic, additive, or synergistic interactions. This analysis compared the observed effects of the drug combinations with the expected effects under the assumption of no interaction, using the highest single agent (HSA) model as a reference. The system assigns a numerical score that classifies the interactions as follows: values below −10 indicate antagonism, scores between −10 and 10 suggest an additive effect, and values above 10 denote synergy. Additionally, a 2D graphical representation was generated, with synergistic regions highlighted in red and antagonistic regions in green, allowing a spatial interpretation of drug interactions. A 3D representation was also included to visualize the dose–response relationship of the triple combination (PTX + DOX + BLM), showing the percentage of cell viability inhibition across different concentrations [72,73].

### 4.5. Assessment of Apoptosis Induction

Apoptosis was analyzed by flow cytometry using Yo-Pro (Invitrogen Co., Carlsbad, CA, USA) and Zombie Aqua (BioLegend, San Diego, CA, USA). Hs-445 cells were treated with PTX, DOX, BLM, or their combinations for 48 h. Briefly, the cells were washed twice with PBS, resuspended in 100 µL of Zombie Aqua (1:20 dilution), and incubated for 20 min at 20 °C, shielded from light. An additional rinse with PBS was performed, and 100 µL of Yo-Pro (1:10,000 dilution) was added and incubated for 10 min under the same conditions. Finally, the cells were resuspended in 100 µL of PBS and analyzed in the CytoFLEX™ flow cytometer (Beckman Coulter, Brea, CA, USA). At least 20,000 events were acquired per sample, and the data were analyzed with Kaluza V2.1 software (Beckman Coulter). Live cells were negative for both dyes; early-apoptotic cells were positive for Yo-Pro but negative for Zombie Aqua; late-apoptotic cells were positive for both dyes; and necrotic cells were positive for Zombie Aqua but negative for Yo-Pro.

### 4.6. Caspase-3, -8, and -9 Activity Assay

Caspase activity was assessed using cleaved caspase-3, -8, and -9 staining kits (Abcam, Cambridge, UK) according to the manufacturer’s instructions. Hs-445 cells were seeded in 24-well plates and treated with PTX, DOX, BLM, or their combinations for 48 h. The cells were harvested, rinsed with PBS, stained with 0.20 µL of FITC-DEVD-FMK, FITC-IETD-FMK, or FITC-LEHD-FMK (1:5 dilution) for caspase-3, -8, -9, respectively, and incubated for 1 h at 37 °C. The samples were washed twice with PBS and resuspended in 250 µL of Wash Buffer. At least 20,000 events per sample were acquired using the CytoFLEX™ flow cytometer (Beckman Coulter). Data were analyzed using Kaluza V2.1 software (Beckman Coulter). Results were expressed as the percentage of cells positive for each cleaved caspase.

### 4.7. Mitochondrial Membrane Potential Assay

To determine ΔΨm, the JC-10 mitochondrial membrane potential assay kit (Abcam, Cambridge, UK) was utilized. Hs-445 cells were treated for 48 h with PTX, DOX, BLM, or their respective combinations. Briefly, cells were rinsed twice with 1X PBS, resuspended in 250 µL of JC-10 dye-loading solution (1:200 dilution in assay buffer) along with 250 µL of 1X PBS, and incubated for 30 min at room temperature, shielded from light. The samples were then processed using the CytoFLEX™ cytometer (Beckman Coulter), with a minimal acquisition of 20,000 events per sample. Data were analyzed with Kaluza V2.1 software (Beckman Coulter). The fluorescence from the monomeric green form of JC-10 was analyzed in the FL1 channel, while the red fluorescence of J-aggregates was analyzed in the FL2 channel.

### 4.8. Cell Cycle Analysis

The Tali Cell Cycle Kit (Invitrogen Co., Carlsbad, CA, USA) was employed to analyze the cell cycle following the manufacturer’s specifications. Hs-445 cells were treated with either PTX, DOX, BLM or their combinations for 48 h. Cells were washed twice with PBS and fixed with 200 µL of 70% ethanol overnight at −20 °C. Following fixation, the cells were washed twice with PBS, stained with 200 µL of the Tali reagent (propidium iodide, RNase A, and Triton X-100), and incubated for 30 min at room temperature, shielded from light. The CytoFLEX™ flow cytometer (Beckman Coulter) was used for analysis, acquiring at least 20,000 events per sample. Data were processed with Kaluza V2.1 software (Beckman Coulter). The results were plotted as the percentage of cells in the G1, S, and G2 phases.

### 4.9. Cell Proliferation Assay

Cell proliferation was measured using the EdU proliferation kit (iFluor 647) (Abcam, Cambridge, UK) according to the manufacturer’s protocol. The Hs-445 cell line underwent treatment with PTX, DOX, BLM, or their combinations for 48 h. Four h prior to the end of treatment, cells were labeled with EdU at a concentration of 15 µM per well. After 48 h of treatment, the cells were harvested and washed with PBS and wash buffer (3% BSA solution in PBS). Subsequently, cells were incubated with 100 µL of 1X Fixative Solution for 20 min at room temperature in the dark. Afterward, the fixed cells were washed twice with wash buffer, resuspended in 100 µL of 1X permeabilization buffer, and incubated for 30 min at room temperature. Next, 500 µL of the reaction mix (438 µL 1X TBS, 10 µL copper sulfate, 2 µL iFluor 647, and 50 µL 1X EdU additive solution) was added to each sample and incubated for 30 min at room temperature, protected from light. After two washes with 1X permeabilization buffer, the cells were resuspended in 300 µL of the same buffer. Samples were analyzed using the CytoFLEX™ cytometer (Beckman Coulter) with the FL4 detector. At least 20,000 events per sample were acquired. The data were analyzed with FlowJo v10 software (BD Biosciences, Franklin Lakes, NJ, USA).

### 4.10. Senescence Assessment

The Senescence Assay Kit (Beta Galactosidase, Fluorescence) (Abcam, Cambridge, UK) was utilized to evaluate cellular senescence. The Hs-445 cell line was treated with PTX, DOX, BLM, or their combinations for 48 h. The cells were harvested, washed with 1X PBS, and incubated with 0.5 µL of senescence reagent in 500 µL of 1X PBS per sample for 20 min at 37 °C. The cells were washed twice with the wash/assay buffer and resuspended in 500 µL of the same buffer. Samples were analyzed using the CytoFLEX™ cytometer (Beckman Coulter), with a minimal acquisition of 20,000 events per sample. Data analysis was carried out using FlowJo v10 software (BD Biosciences).

### 4.11. Clonogenic Capacity Assessment

To evaluate the effects of PTX, DOX, BLM, and their combinations on the clonogenic capacity of Hs-445 cells, viable cells were isolated by cell sorting after 48 h of treatment. Subsequently, the Hematopoietic Colony Forming Cell Assay kit (Abcam, Cambridge, UK) was used, which allows the quantification of the total DNA in a sample. This value is proportional to the number of cells through cyanine-based fluorescence detection.

Initially, treated cells were stained with 100 µL of Sytox (1:1000 dilution) and 100 µL of Yo-Pro (1:10,000 dilution), incubated for 15 min at 25 °C protected from light. Next, 100,000 viable cells (double negative to Sytox and Yo-Pro) were sorted using a FACSAria III cell sorter (BD Biosciences^®^). From each experimental group, 5000 viable cells were seeded into 96-well plates. Subsequently, 135 µL of methylcellulose was added to each well, and the mixture was homogenized. Plates were incubated for 10 days at 37 °C with 5% CO_2_ atmosphere. Following the incubation period, 50 µL/well of the lysis solution/dye-GR (1:75 dilution) was added and incubated for 30 min at 25 °C. Finally, 100 µL from each well was transferred to a new black-bottomed 96-well plate suitable for fluorescence detection, which was performed at 480 nm using a Synergy™ HT plate reader (Biotek, Winooski, VT, USA). The results were expressed as relative fluorescence units (RFU).

### 4.12. Assessment of Gene Expression by qPCR

To evaluate gene expression, 5 × 10^6^ cells were seeded in Petri dishes and treated for 6 h with PTX, DOX, BLM, or their combinations under the previously described culture conditions. Total RNA was extracted from each experimental group using the Quick-RNA Miniprep Plus Kit (Zymo Research, CA, USA), following the manufacturer’s specifications. Subsequently, cDNA was synthesized from 5 µg of total RNA per group using the SCRIPT cDNA Synthesis Kit (Jena Bioscience, Jena, Germany).

Quantitative polymerase chain reaction (qPCR) was performed using the CFX Opus 96 Real-Time PCR System (BIO-RAD, Hercules, CA, USA) and the LightCycler^®^ FastStart DNA MasterPLUS SYBR Green I kit (Roche, Mannheim, Germany). The PCR program included a 10 min pre-incubation at 95 °C, followed by 40 amplification cycles consisting of three steps: denaturation at 95 °C for 10 s, annealing for 10 s at a primer-specific temperature (Table 2), and extension at 72 °C for 15 s. All samples were processed in duplicate. The housekeeping genes RPS18 and RPLP10 were used to normalize gene expression levels.

PCR products were analyzed with the CFX Maestro 2.3 Software (BIO-RAD), and relative expression levels were calculated using the 2^−ΔΔCp^ method and expressed as Log_2_ Fold Change (Log2FC). Oligonucleotides were designed using Oligo v6 software based on sequences available in the GenBank database of the National Center for Biotechnology Information (http://www.ncbi.nlm.nih.gov, accessed on 17 March 2024) (Table 2).

### 4.13. Statistical Analysis

The results of each assay are expressed as the mean ± standard deviation (SD) from three independent experiments conducted in duplicate. Statistical analysis was performed using the Student’s *t*-test and two-way ANOVA for parametric data, followed by the Dunnett post hoc test. For the PCR analysis, one-way ANOVA and Tukey post hoc test were used, comparing ΔCp values among treatment groups for each gene analyzed. Analyses were carried out using GraphPad Prism ver. 8 statistical software. A *p*-value < 0.05 was considered statistically significant.

## 5. Conclusions

Overall, PTX emerges as a promising candidate for the treatment of HL due to its significant antitumor activity. PTX induces tumor apoptosis alone and robustly potentiates DOX- and BLM-induced cytotoxicity, generating synergistic interactions. In addition, PTX reduces DOX- and BLM-induced senescence. This effect favors the efficient elimination of tumor cells, decreasing the proportion of cells with a senescent phenotype that contributes to a proinflammatory tumor microenvironment. PTX also enhances the reduction in colony-forming capacity exerted by DOX and BLM, thereby decreasing the probability of the survival of treatment-resistant cells with long-term proliferation potential.

In this context, the combination of PTX + DOX + BLM demonstrates significant clinical potential for treating HL by effectively eliminating tumor cells. Additionally, PTX has the potential to reduce the severity and frequency of chemotherapy-associated adverse effects, such as DOX-associated cardiotoxicity and BLM-induced pulmonary fibrosis. This strategy could improve treatment efficacy, enhance the patient’s quality of life, and potentially increase survival rates.

Furthermore, PTX improves treatment efficacy in HL by acting on key aspects of tumor biology, such as viability and cell proliferation, and by promoting tumor cell death through various mechanisms, such as caspase activation, mitochondrial membrane depolarization, cell cycle deregulation, upregulation of proapoptotic genes, and downregulation of antiapoptotic genes. However, further research is needed to evaluate its translational potential and clinical efficacy in preclinical studies and clinical trials in HL patients.

The incorporation of PTX into chemotherapy regimens offers new therapeutic strategies to enhance clinical outcomes and mitigate the adverse effects of chemotherapy, thus improving the quality of life and survival rates for patients with HL.

## Figures and Tables

**Figure 1 cimb-47-00593-f001:**
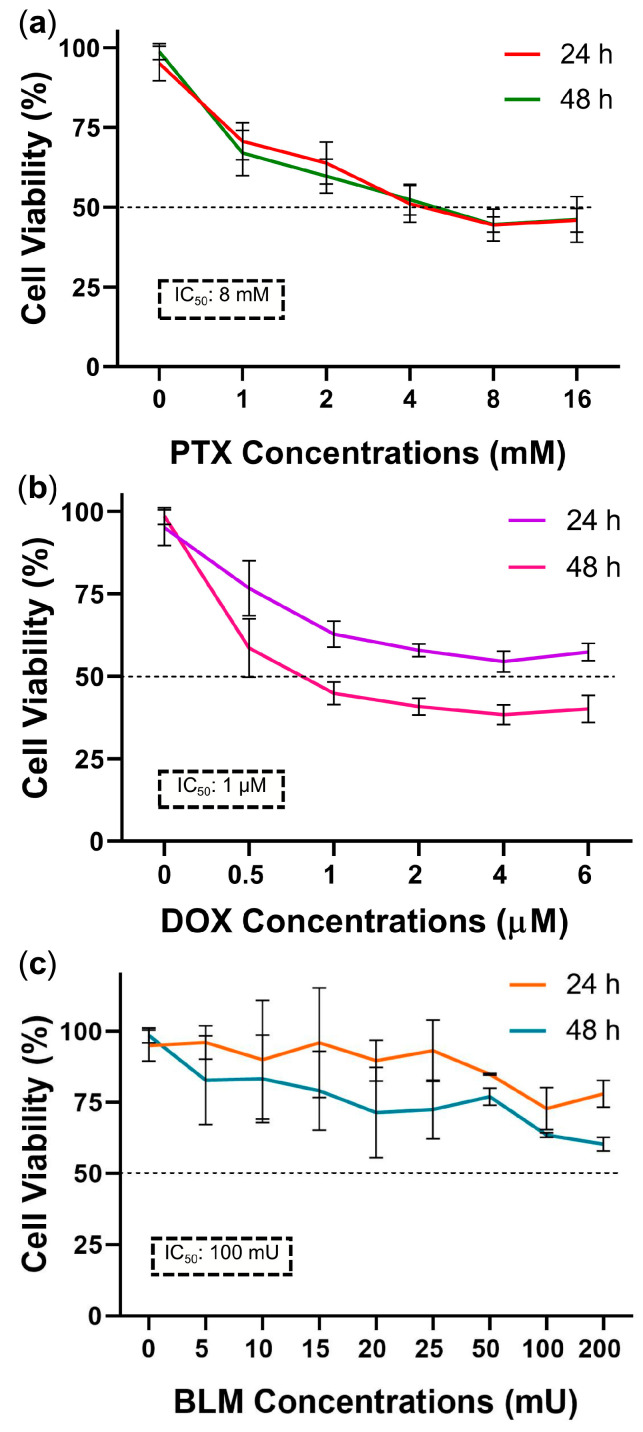
Dose–response Curves for PTX, DOX, and BLM in Hs-445 HL Cells. (**a**–**c**) Dose–response curves of Hs-445 cells treated with (**a**) PTX (1, 2, 4, 8, and 16 mM), (**b**) DOX (0.5, 1, 2, 4, and 6 µM), and (**c**) BLM (5, 10, 15, 20, 25, 50, 100, and 200 mU) for 24 and 48 h. The curves were generated by a cell viability assay utilizing WST-1. The IC_50_ values determined were 8 mM for PTX, 1 µM for DOX, and 100 mU for BLM. Data are presented as the mean ± standard deviation (SD) (n = 3). BLM: bleomycin; DOX: doxorubicin; IC_50_: half-maximal inhibitory concentration; PTX: pentoxifylline.

**Figure 2 cimb-47-00593-f002:**
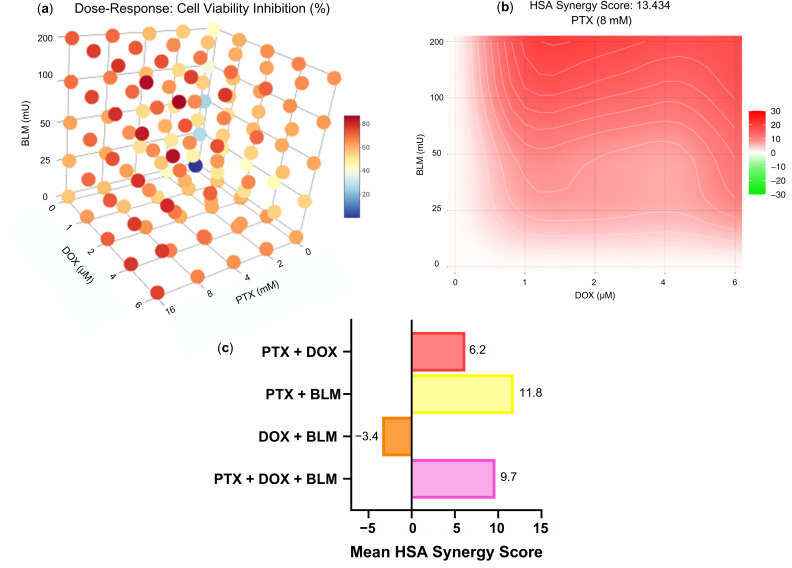
Evaluation of the pharmacological interaction between PTX, DOX, and BLM. The combined effects of PTX + DOX, PTX + BLM, DOX + BLM, and PTX + DOX + BLM were analyzed in Hs-445 cells after 48 h of exposure to determine whether they exhibited synergistic, antagonistic, or additive interactions. The analysis was performed using the SynergyFinder ver. 3.0 tool with the highest single agent (HSA) model as a reference. Synergy scores below −10 indicate antagonism, scores between −10 and 10 reflect an additive effect, and scores above 10 indicate synergy. (**a**) 3D representation of the dose–response inhibition of cell viability (%) for the combinations of DOX (0–6 µM), BLM (0–200 mU), and PTX (0–16 mM). (**b**) 2D graphical representation of the HSA synergy score for the combinations of DOX (0–6 µM) and BLM (0–200 mU) with PTX at a fixed concentration of 8 mM, where antagonistic regions are shown in green and synergistic regions in red. (**c**) The mean HSA synergy scores are presented as bar graphs for each treatment group across all tested concentrations of PTX, DOX, and BLM. BLM: bleomycin; DOX: doxorubicin; HSA: highest single agent model; PTX: pentoxifylline.

**Figure 3 cimb-47-00593-f003:**
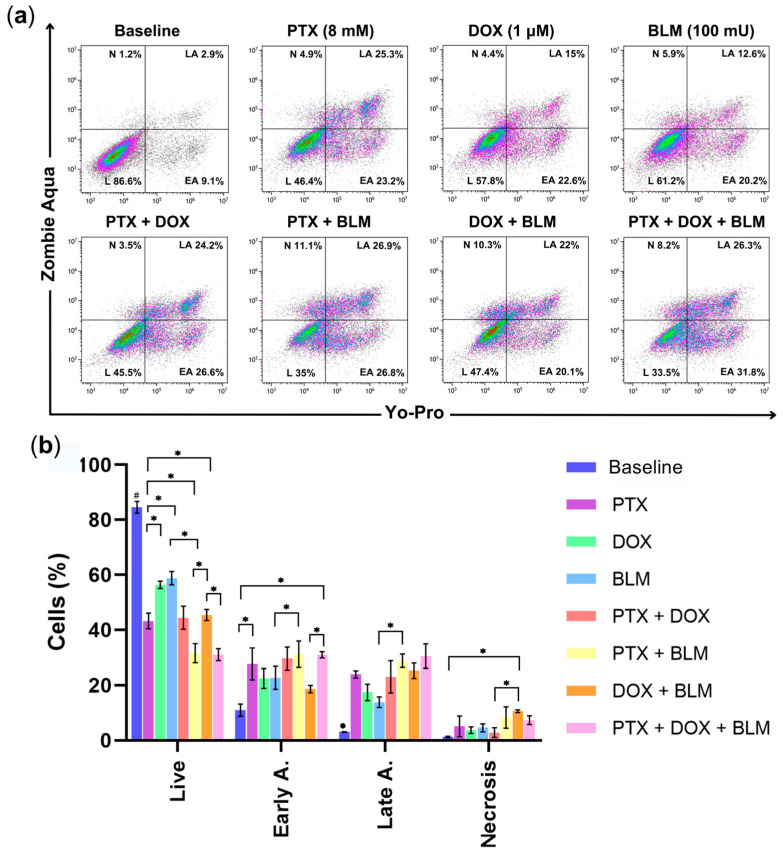
PTX induces apoptosis in Hs-445 cells and increases the apoptosis generated by DOX and BLM. The cells were exposed to the drugs individually and in combinations for 48 h. (**a**) Representative density plots distinguishing live (L), early-apoptotic (EA), late-apoptotic (LA), and necrotic (N) Hs-445 cells after treatment with PTX (8 mM), DOX (1 µM), BLM (100 mU), and their combinations. (**b**) The apoptosis of Hs-445 cells treated with PTX, DOX, BLM, and their combinations are presented as bar charts, displaying the mean ± standard deviation (SD) (n = 3). Two-way ANOVA and Dunnett post hoc tests were used. Statistical significance: # (*p* < 0.05) baseline vs. all treatment groups; * (*p* < 0.05) vs. the different groups compared. Baseline: cells without treatment; PTX: pentoxifylline; DOX: doxorubicin; BLM: bleomycin; L: Live; EA: early apoptosis; LA: late apoptosis; N: necrosis.

**Figure 4 cimb-47-00593-f004:**
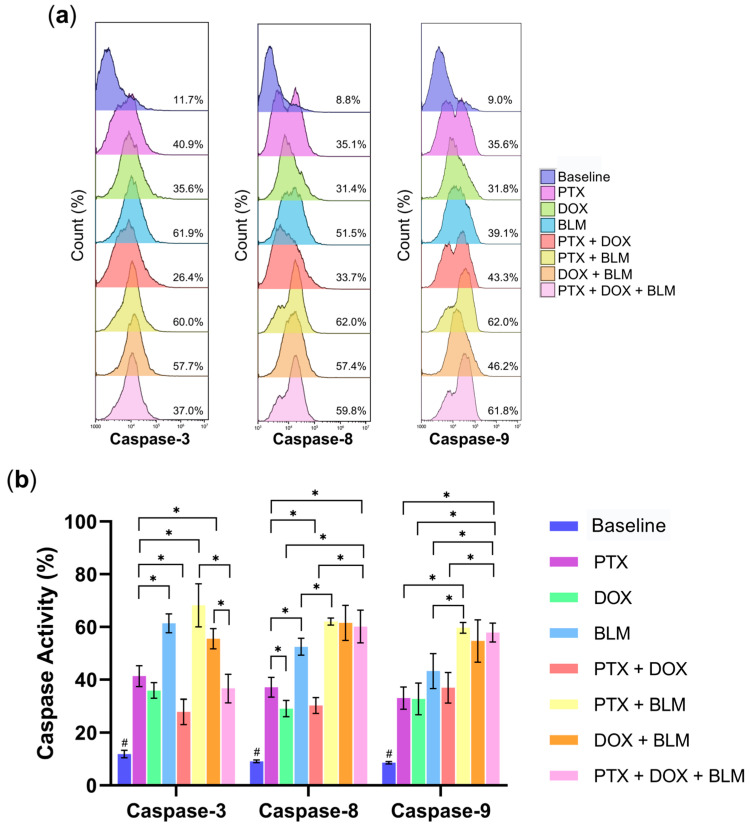
Combined PTX and BLM treatment increases the activity of caspase-3, -8, and -9 in Hs-445 Cells. Caspase activity was evaluated using flow cytometry. Hs-445 cells were treated with PTX (8 mM), DOX (1 µM), BLM (100 mU), and their combinations for 48 h. (**a**) Representative histograms of caspase-3, -8, and -9 activity in Hs-445 cells after treatment with PTX (8 mM), DOX (1 µM), BLM (100 mU), and their combinations. (**b**) The results of caspases-3, -8, and -9 activity in Hs-445 cells treated with PTX, DOX, BLM, and their combinations are presented as bar charts, showing the mean ± standard deviation (SD) (n = 3). Student’s *t*-test was used to assess the results. Statistical significance: # (*p* < 0.05) baseline vs. all treatment groups; * (*p* < 0.05) vs. the different groups compared. Baseline: cells without treatment; PTX: pentoxifylline; DOX: doxorubicin; BLM: bleomycin.

**Figure 5 cimb-47-00593-f005:**
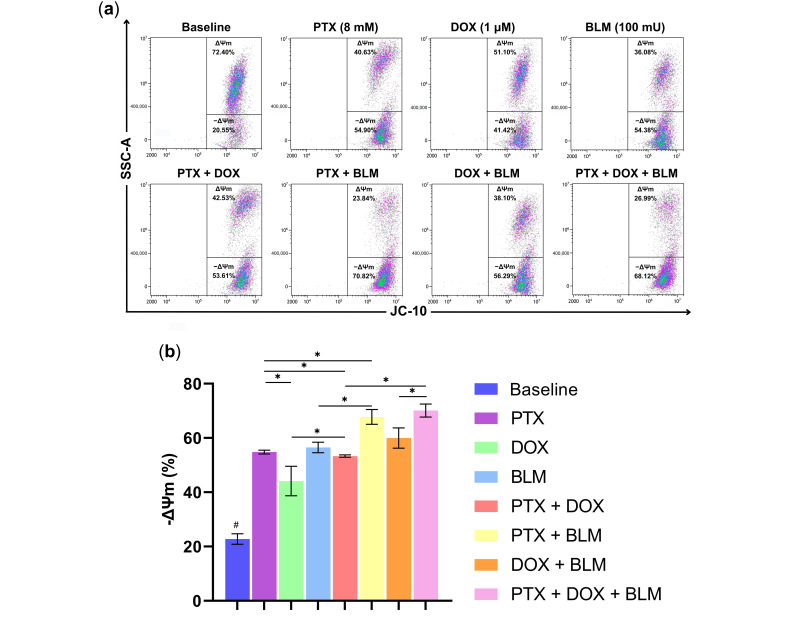
PTX decreases the ΔΨm in Hs-445 Cells, with a pronounced effect observed when combined with BLM. Cells were exposed to PTX (8 mM), DOX (1 µM), BLM (100 mU), and their respective combinations for 48 h. The ΔΨm was determined by flow cytometry. (**a**) Representative density plots of the ΔΨm in Hs-445 cells after treatment with PTX (8 mM), DOX (1 µM), BLM (100 mU), and their combinations. (**b**) Bar charts displaying the percentage of Hs-445 cells with mitochondrial depolarization after treatment with PTX, DOX, BLM, and their combinations, presented as the mean ± standard deviation SD (n = 3). Results were analyzed using Student’s *t*-test. Statistical significance: # (*p* < 0.05) baseline vs. all treatment groups; * (*p* < 0.05) vs. the different groups compared. Baseline: untreated cells; PTX: pentoxifylline; DOX: doxorubicin; BLM: bleomycin; ΔΨm: mitochondrial membrane potential.

**Figure 6 cimb-47-00593-f006:**
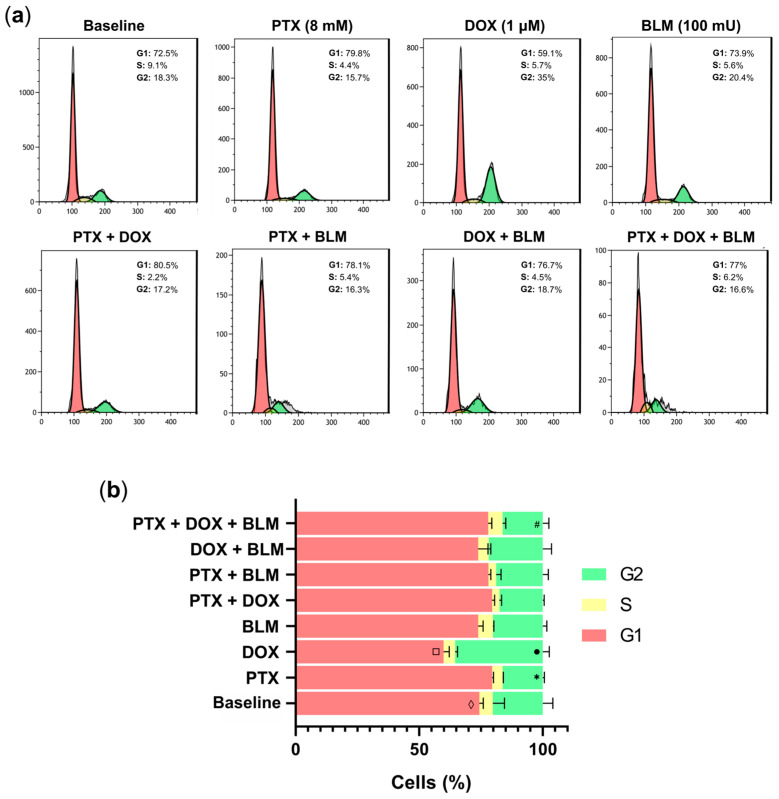
PTX abrogates DOX-induced cell blockade in the G2 phase and, in turn, increases the percentage of cells in the G1 Phase in Hs-445 HL cells. The progression of the cell cycle was determined using flow cytometry. (**a**) Representative histograms depicting the percentage of Hs-445 cells treated with PTX (8 mM), DOX (1 µM), BLM (100 mU), and their combinations for 48 h, identified in each cell cycle phase: G1, S, and G2. (**b**) Bar charts illustrating the percentage of Hs-445 cells in each cell cycle phase after treatment with PTX, DOX, BLM, and their combinations. The results display the mean ± standard deviation (SD) (n = 3). Two-way ANOVA and Dunnett post hoc test were used. Statistical significance: □ (*p* < 0.05) DOX vs. all treatment groups in the G1 phase; ◊ (*p* < 0.05) Baseline vs. PTX + DOX + BLM in the G1 phase; • (*p* < 0.05) DOX vs. all treatment groups in the G2 phase; * (*p* < 0.05) PTX vs. PTX + DOX in the G2 phase; # (*p* < 0.05) PTX + DOX + BLM vs. BLM in the G2 phase. Baseline: cells without treatment; PTX: pentoxifylline; DOX: doxorubicin; BLM: bleomycin.

**Figure 7 cimb-47-00593-f007:**
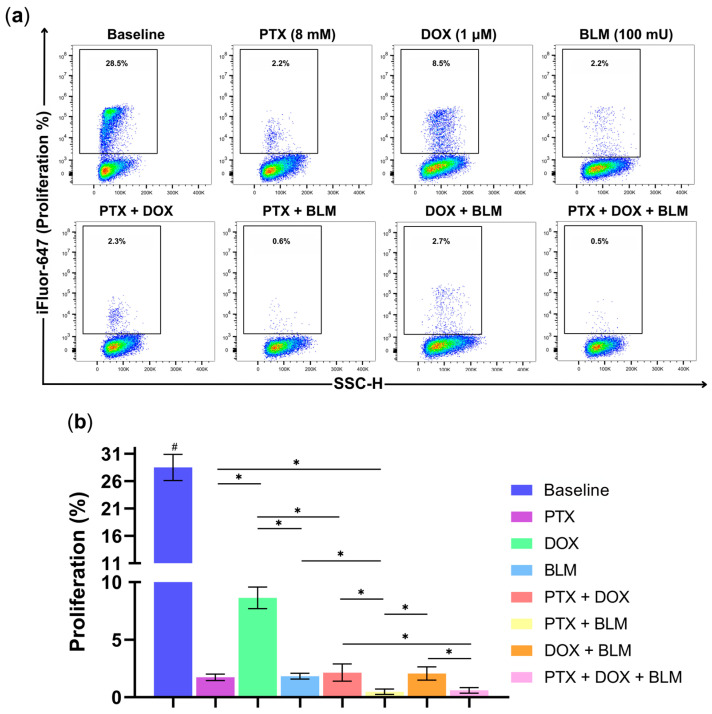
PTX decreases the proliferation of Hs-445 HL tumor cells. Cells were treated with PTX (8 mM), DOX (1 µM), BLM (100 mU), and their respective combinations for 48 h. The proliferative capacity of the cells was assessed by flow cytometry. (**a**) Representative density plots of Hs-445 cell proliferation after treatment with PTX (8 mM), DOX (1 µM), BLM (100 mU), and their combinations. (**b**) Bar charts displaying the percentage of proliferation in Hs-445 cells treated with PTX, DOX, BLM, and their combinations, presented as the mean ± standard deviation SD (n = 3). Student’s *t*-test was used to analyze the data. Statistical significance: # (*p* < 0.05) baseline vs. all treatment groups; * (*p* < 0.05) vs. the different groups compared. Baseline: untreated cells; PTX: pentoxifylline; DOX: doxorubicin; BLM: bleomycin.

**Figure 8 cimb-47-00593-f008:**
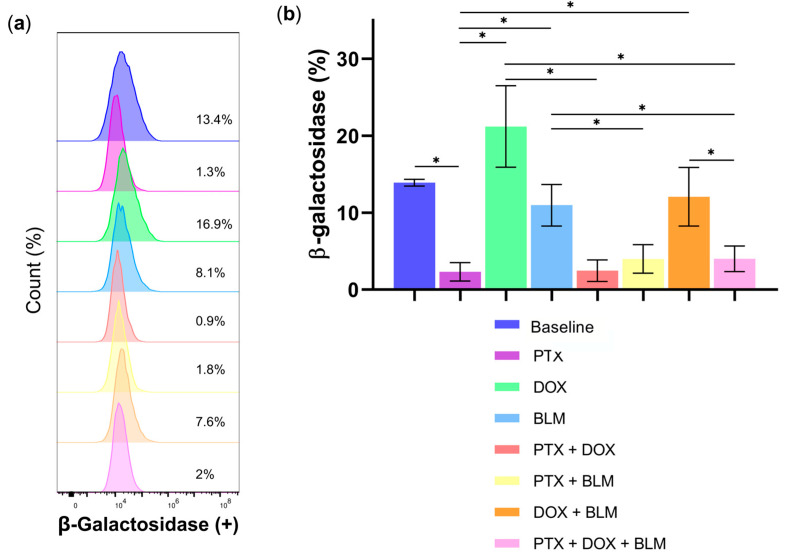
PTX Inhibits DOX- and BLM-induced Senescence in Hs-445 HL Cells. Cells were exposed to PTX (8 mM), DOX (1 µM), BLM (100 mU), and their combinations for 48 h. Senescence was assessed by measuring β-galactosidase. (**a**) Representative histogram of senescence analysis by flow cytometry in Hs-445 cells treated with PTX (8 mM), DOX (1 µM), BLM (100 mU), and their combinations. (**b**) Bar charts exhibiting the percentage of senescent Hs-445 cells after treatment with PTX, DOX, BLM, and their combinations, presented as the mean ± standard deviation SD (n = 3). Results were analyzed with Student’s *t*-test. Statistical significance: * (*p* < 0.05) vs. the different groups compared. Baseline: untreated cells; PTX: pentoxifylline; DOX: doxorubicin; BLM: bleomycin.

**Figure 9 cimb-47-00593-f009:**
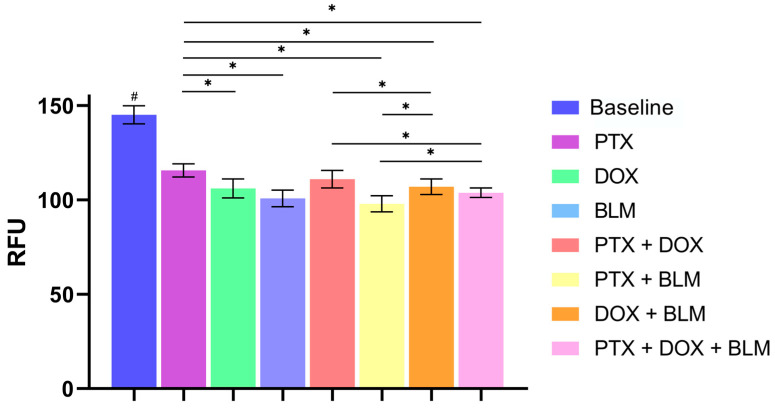
PTX and BLM treatment decreases the clonogenic activity of HL cells. The clonogenic capacity was evaluated by cell sorting followed by fluorescence detection. Hs-445 cells were treated with PTX (8 mM), DOX (1 µM), BLM (100 mU), and their combinations for 48 h. Viable cells were then sorted and incubated in methylcellulose for 10 days to assess colony-forming ability by fluorescence detection. The figure shows a representative histogram of clonogenic capacity in Hs-445 cells. Results are presented as bar charts showing mean RFU ± standard deviation (SD) (n = 3). Statistical analysis was performed using Student’s *t*-test. Statistical significance: # (*p* < 0.05) baseline vs. all treatment groups; * (*p* < 0.05) vs. the different groups compared. Baseline: cells without treatment; PTX: pentoxifylline; DOX: doxorubicin; BLM: bleomycin.

**Figure 10 cimb-47-00593-f010:**
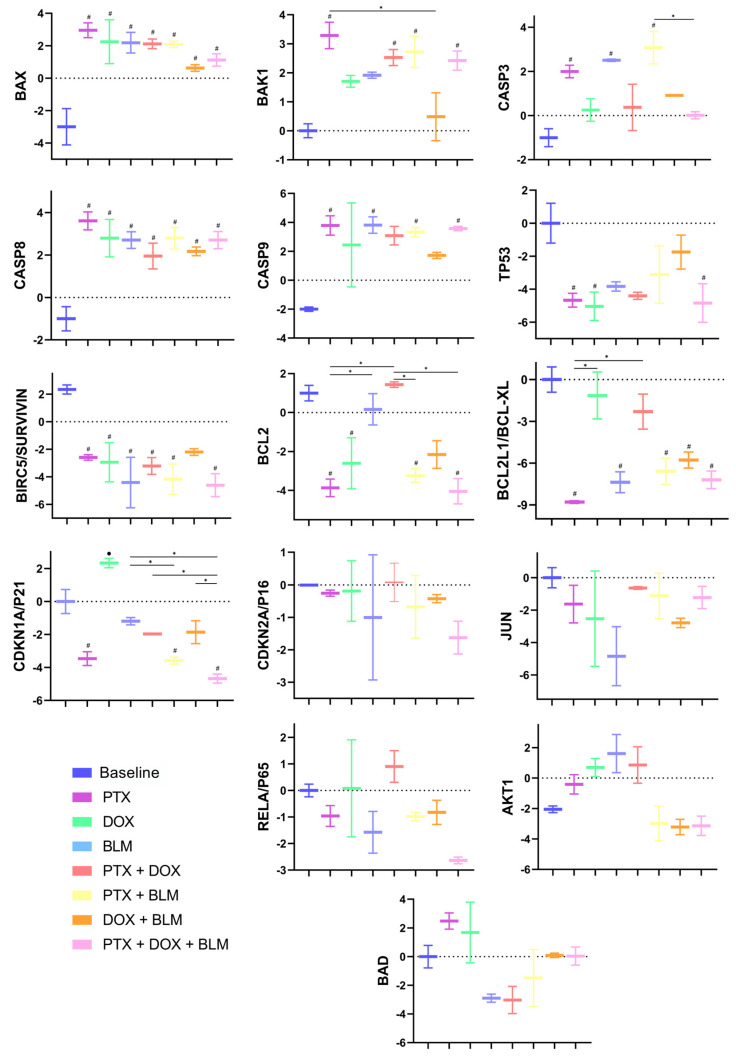
Effect of treatment with PTX, DOX, BLM, and their combinations on the expression of genes related to apoptosis, senescence, and signaling pathways in HL cells. To evaluate the effect of treatments on the expression of genes associated with apoptosis, senescence, and signaling pathways, Hs-445 cells were treated for 6 h with PTX (8 mM), DOX (1 µM), BLM (100 mU), and their combinations. Gene expression was analyzed by qPCR using the CFX Maestro 2.3 software. Relative expression levels were calculated using the 2^−ΔΔCp^ method and expressed as Log_2_ Fold Change (Log2FC). Results are presented as bar charts showing the mean ± minimum and maximum values, with RPS18 and RPLP10 used as reference genes for normalization (n = 2). Statistical analysis was performed using one-way ANOVA and Tukey post hoc test, comparing ΔCp values among treatment groups for each gene analyzed. Statistical significance: # (*p* < 0.05) vs. the baseline group; • (*p* < 0.05) DOX group vs. the different groups; * (*p* < 0.05) vs. the different groups compared. Baseline: cells without treatment; PTX: pentoxifylline; DOX: doxorubicin; BLM: bleomycin.

**Table 1 cimb-47-00593-t001:** Chemical Structures of bleomycin, doxorubicin, and pentoxifylline.

Name and Abbreviation	Molecular Formula	Chemical Structure	PubChem CID
Bleomycin (BLM)	C_60_H_96_N_20_O_21_S_2_	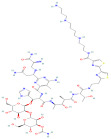	9877229
Doxorubicin (DOX)	C_27_H_29_NO_11B_	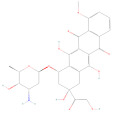	31703
Pentoxifylline (PTX)	C_13_H_18_N_4_O_3_	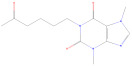	4740

CID: Compound Identifier.

**Table 2 cimb-47-00593-t002:** Primers sequences.

Gene	Primer Sequence	Amplicon Length	Annealing Temperature	GenBank Accession No.
*BAD*	F: 5′ CTC CGG AGG ATG AGT CAC GAG T 3′ R: 5′ ACT TCC GCC CAT ATT CAA GAT 3′	240 bp	60 °C	NM_004322.2
*BCL2L1* *(BCL-XL)*	F: 5′ GCA GGC GAC GAG TTT GAA CT 3′ R: 5′ GTG TCT GGT CAT TTC CGA CTG A 3′	434 bp	60 °C	NM_138578.3
*BCL2*	F: 5′ CGA CTT CTC CCG CCG CTA CC 3′ R: 5′ CCG CAT GCT GGG GCC GTA CAG 3′	316 bp	60 °C	NM_000633.3
*BIRC5* *(SURVIVIN)*	F: 5′ TGA GCT GCA GGT TCC TTA TCT G 3′ R: 5′ GAA TGG CTT TGT GCT TAG TTT T 3′	234 bp	59 °C	NM_001168.3
*CASP3*	F: 5′ ATA CTC CAC AGC ACC TGG TTA T 3′ R: 5′ AAT GAG AGG GAA ATA CAG TAC CAA 3′	329 bp	62 °C	NM_004346.4
*CASP8*	F: 5′ ACC TGC TGG ATA TTT TCA TAG AGA 3′ R: 5′ TGT TGA TGA TCA GAC AGT ATC CC 3′	264 bp	65 °C	NM_001228.5
*CASP9*	F: 5′ GTA CGT TGA GAC CCT GGA CGA C 3′ R: 5′ GCT GCT AAG AGC CTG TCT GTC ACT 3′	323 bp	65 °C	NM_001229.5
*RELA* *(P65)*	F: 5′ GCA GGC TCC TGT GCG TGT CT 3′ R: 5′ GGT GCT CAG GGA TGA CGT AAA G 3′	286 bp	60 °C	NM_021975.4
*TP53* *(P53)*	F: 5′ CTG AGG TTG GCT CTG ACT GTA CCA CCA TCC 3′ R: 5′ CTC ATT CAG CTC TCG GAA CAT CTC GAA GCG 3′	371 bp	60 °C	NM_000546.6
*CDKN1A* *(P21)*	F: 5′ CGA CTT TGT CAC CGA GAC AC 3′ R: 5′ CGT TTT CGA CCC TGA GAG T 3′	273 bp	60 °C	NM_000389.5
*BAX*	F: 5′ TTT GCT TCA GGG TTT CAT CC 3′ R: 5′ CAG TTG AAG TTG CCG TCA GA 3′	246 bp	60 °C	NM_001291428.2
*BAK1*	F: 5′ CGC TTC GTG GTC GAC TTC AT 3′ R: 5′ AGA AGG CAA AGA CTT CGC TTA 3′	240 bp	60 °C	NM_001188.3
*JUN*	F: 5′ TGG AAA GTA CTC CCC TAA CCT 3′ R: 5′ CTG AAA CAT CGC ACT ATC CTT 3′	250 bp	58 °C	NM_002228.3
*AKT1*	F: 5′ TCA CGT CGG AGA CTG ACA 3′ R: 5′ AAA TAC AGA TCA TGG CAC GAG 3′	229 bp	60 °C	NM_005163.2
*CDKN2A* *(P16)*	F: 5′ GGC ACC AGA GGC AGT AAC C 3′ R: 5′ CTA CGA AAG CGG GGT GGG 3′	172 bp	60 °C	NM_000077.5
*RPS18*	F: 5′ CGA TGG GCG GCG GAA AA 3′ R: 5′ CAG TCG CTC CAG GTC TTC ACG G 3′	283 bp	58 °C	NM_022551.2
*RPLP0*	F: 5′ CCT CAT ATC CGG GGG AAT GTG 3′ R: 5′ GCA GCA GCT GGC ACC TTA TTG 3′	95 bp	60 °C	NM_001002.4

## Data Availability

The data presented in this work will be made available upon request.

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
