# Peer review of "Pentoxifylline Enhances the Effects of Doxorubicin and Bleomycin on Apoptosis, Caspase Activity, and Cell Cycle While Reducing Proliferation and Senescence in Hodgkin’s Disease Cell Line"

_cimb, 2025, doi:10.3390/cimb47080593_

Round 1
Reviewer 1 Report
Comments and Suggestions for Authors
Overall, this is a solid paper. Here are some comments to improve the paper.
- The finding that PTX reverses DOX/BLM-induced senescence in Hs-445 cells is highly significant and represents a novel contribution. While PTX's pro-apoptotic and synergistic effects with chemotherapeutics are valuable, the observation that it counters therapy-induced senescence (TIS) is particularly intriguing. TIS can promote a pro-inflammatory, pro-tumorigenic microenvironment and contribute to relapse. The authors convincingly demonstrate PTX's ability to shift cells from senescence towards apoptosis using robust assays (flow cytometry for SA-β-gal, apoptosis, proliferation). This mechanistic insight into PTX potentially mitigating an adverse consequence of chemotherapy (persistent senescent cells) strengthens its rationale as a combination partner. The discussion effectively links this finding to potential clinical benefits (reduced TME risk). Further elaboration on the specific mechanisms by which PTX achieves this senescence reversal (beyond general caspase activation/ΔΨm loss) would be a valuable future direction.
- The manuscript repeatedly states 'synergistic' effects for PTX combinations (especially triple therapy) based on enhanced effects compared to single agents. However, a more rigorous quantitative assessment of synergy (e.g., calculating Combination Index (CI) using Chou-Talalay or similar methods, or generating isobolograms) is lacking. While the data (cell viability, clonogenic survival, apoptosis, ΔΨm) strongly suggest synergy, definitive proof requires formal synergy analysis. Providing CI values for key combinations/doses would significantly strengthen the claims of synergy and allow better comparison with other studies. Furthermore, details on the specific concentrations used for each drug in the combination experiments (beyond stating they were used at IC50 or EC50) are crucial for reproducibility. A table summarizing the exact concentrations used for single agents and combinations in each major assay (viability, clonogenic, apoptosis, cell cycle, senescence) would greatly enhance methodological clarity.
- The translational implications of PTX reducing chemotherapy side effects (cardiotoxicity, pulmonary fibrosis) are compellingly discussed. However, these potential benefits are inferred solely from in vitro cytotoxicity/senescence data on one cell line (Hs-445). Direct evidence that PTX protects normal tissues from DOX/BLM toxicity is absent from this study. While beyond the current scope, acknowledging this limitation explicitly is important. Furthermore, the reliance on a single HL cell line is a significant constraint. Demonstrating key findings (especially the synergistic apoptosis induction and senescence reversal) in additional HL cell lines (e.g., L428, KM-H2) or, ideally, primary HL samples would greatly strengthen the generalizability and impact of the conclusions. The potential impact of PTX on normal lymphocytes should also be briefly discussed to contextualize potential clinical side effects. The call for future in vivo and clinical studies is appropriate and necessary to validate these promising in vitro results.
Author Response
Comments 1: The finding that PTX reverses DOX/BLM-induced senescence in Hs-445 cells is highly significant and represents a novel contribution. While PTX's pro-apoptotic and synergistic effects with chemotherapeutics are valuable, the observation that it counters therapy-induced senescence (TIS) is particularly intriguing. TIS can promote a pro-inflammatory, pro-tumorigenic microenvironment and contribute to relapse. The authors convincingly demonstrate PTX's ability to shift cells from senescence towards apoptosis using robust assays (flow cytometry for SA-β-gal, apoptosis, proliferation). This mechanistic insight into PTX potentially mitigating an adverse consequence of chemotherapy (persistent senescent cells) strengthens its rationale as a combination partner. The discussion effectively links this finding to potential clinical benefits (reduced TME risk). Further elaboration on the specific mechanisms by which PTX achieves this senescence reversal (beyond general caspase activation/ΔΨm loss) would be a valuable future direction.
Response 1: We fully agree with your comment. Indeed, delving into the specific mechanisms that enable the reversal of the senescent phenotype is essential for a clearer understanding of the mechanism of action of pentoxifylline (PTX), not only in the context of cellular senescence, but also concerning its broader antitumor effects on cancer cells.
Several studies have documented that pentoxifylline exerts a direct effect on cell cycle modulation, particularly through the regulation of the cell cycle inhibitors p21 and p16. In the present study, although no significant changes were observed in the expression of the CDKN2A/p16 gene in our experimental model, a significant reduction in CDKN1A/p21 expression, one of the key markers associated with the senescent phenotype, was detected. This finding becomes relevant considering that treatment with doxorubicin (DOX) markedly increased CDKN1A/p21 expression, in line with its known prosenescent effect.
It was also observed that PTX also reduced the expression of TP53, a tumor suppressor gene with proapoptotic functions that also plays a role in regulating cellular senescence. This reduction in TP53 expression could contribute to the inhibition of the senescent phenotype in Hodgkin Lymphoma (HL) cells, beyond its direct involvement in apoptosis, especially considering that this effect is accompanied by upregulation of other proapoptotic genes.
These observations suggest a possible mechanism by which PTX may reverse senescence, beyond the general activation of caspases and the decrease in mitochondrial membrane potential. However, further studies are required to elucidate the pathways involved in this process more precisely. (Discussion: p. 18, lines 588-604).
For example, it would be relevant to explore the potential involvement of pentoxifylline in the regulation of proteins such as Mcl-1, which has been shown to play a role in the establishment of tumor cell senescence (reference new article), or in the modulation of various signaling pathways such as mTOR, components of the MAPK pathway, or heat shock proteins like HSP90, all of which are strongly implicated in the regulation of the senescence-associated secretory phenotype (SASP).
Comments 2: The manuscript repeatedly states 'synergistic' effects for PTX combinations (especially triple therapy) based on enhanced effects compared to single agents. However, a more rigorous quantitative assessment of synergy (e.g., calculating Combination Index (CI) using Chou-Talalay or similar methods, or generating isobolograms) is lacking. While the data (cell viability, clonogenic survival, apoptosis, ΔΨm) strongly suggest synergy, definitive proof requires formal synergy analysis. Providing CI values for key combinations/doses would significantly strengthen the claims of synergy and allow better comparison with other studies. Furthermore, details on the specific concentrations used for each drug in the combination experiments (beyond stating they were used at IC50 or EC50) are crucial for reproducibility. A table summarizing the exact concentrations used for single agents and combinations in each major assay (viability, clonogenic, apoptosis, cell cycle, senescence) would greatly enhance methodological clarity.
Response 2: Thank you for pointing this out. The analysis of the pharmacological effects of the different drug combinations (PTX + DOX, PTX + BLM, DOX + BLM, and PTX + DOX + BLM) was based on the results of the cell viability assay, as well as on the dose–response curves obtained for each individual agent using the WST-1 reagent. Subsequently, the SynergyFinder web platform was used to evaluate whether the experimental combinations of PTX, DOX, and BLM exhibited antagonistic, additive, or synergistic interactions.
This software enables the mathematical calculation of the expected effects of drug combinations using various reference models or null models, which quantify interactions
under the assumption of no synergy between the individual compounds. SynergyFinder facilitates the simultaneous analysis and interactive visualization of drug combinations through multiple mathematical models, allowing the analysis to be tailored to the characteristics of the experimental data. This multi-model, multi-sample strategy considers the contextual dependence of drug interactions, minimizes the risk of false positives due to experimental errors, and provides a more robust interpretation of combination effects.
The platform implements several reference models, including the Bliss independence model, Loewe additivity, the Highest Single Agent (HSA) model, and the Zero Interaction Potency (ZIP) model. In this study, the model that best fit our data was the HSA model, as the dose–response curves of certain drugs (e.g., BLM) did not exhibit linear behavior. Additionally, some models require detailed mechanistic knowledge of the drugs' actions, which is not fully available for PTX.
The HSA model considers that a positive (synergistic) interaction exists when the response obtained by the combination (EAB) is greater than that obtained with the most active drug separately (EAB > max(EA, EB)). Under this model, the software generates a combination index (CI) based on the following formula: CI = max(EA, EB)/EAB, interpreted as follows: values less than –10 indicate antagonism, values between –10 and 10 indicate additive effects, and values greater than 10 indicate synergy. Although this index serves a similar purpose to the CI proposed by the Chou–Talalay method, its values do not follow the same criteria of >1, =1, or <1, as the SynergyFinder CI depends on the mathematical model applied (in this case, HSA).
In section 4.3. Cell Culture and Experimental Conditions of the Materials and Methods section, specific details of the concentrations used for each drug in the viability assay to determine the IC50 and the analysis of the pharmacological effects have been incorporated: PTX (1, 2, 4, 4, 8 and 16 mM), DOX (0.5, 1, 1, 2, 4 and 6 μM), and BLM (5, 10, 15, 20, 25, 50, 100 and 200 mU). Additionally, a database containing the CI values obtained for each analyzed combination has been included in the supplementary material.
Finally, the same section specifies the final concentrations selected for experiments involving individual and combined treatments: PTX (4 mM), DOX (1 µM), and BLM (100 mU). These concentrations were used consistently throughout the study (Materials and Methods: p. 20, lines 664-672; Results: p. 4, lines 152–153).
Comments 3: The translational implications of PTX reducing chemotherapy side effects (cardiotoxicity, pulmonary fibrosis) are compellingly discussed. However, these potential benefits are inferred solely from in vitro cytotoxicity/senescence data on one cell line (Hs-445). Direct evidence that PTX protects normal tissues from DOX/BLM toxicity is absent from this study. While beyond the current scope, acknowledging this limitation explicitly is important. Furthermore, the reliance on a single HL cell line is a significant constraint. Demonstrating key findings (especially the synergistic apoptosis induction and senescence reversal) in additional HL cell lines (e.g., L428, KM-H2) or, ideally, primary HL samples would greatly strengthen the generalizability and impact of the conclusions. The potential impact of PTX on normal lymphocytes should also be briefly discussed to contextualize potential clinical side effects. The call for future in vivo and clinical studies is appropriate and necessary to validate these promising in vitro results.
Response 3: We appreciate your comment, which we acknowledge and agree with.
There is evidence demonstrating the protective effect of PTX on healthy tissues, as it does not induce cytotoxicity in non-tumor cells. For example, in several studies evaluating the effect of PTX in combination with cisplatin or adriamycin in cervical cancer cell lines, a non-tumor epithelial line (HaCaT) was also used. In these studies, various concentrations of PTX (1, 2, 4, 8, 16, and 20 mM) had no significant effect on cell viability compared to untreated controls, regardless of the administered dose. Moreover, no significant differences were observed in early or late apoptosis, caspase activation, or in the expression of proteins such as p-p38 and p-p65 following PTX treatment.
Similarly, when analyzing gene expression of various apoptosis-related markers, exclusive PTX treatment did not induce overexpression of proapoptotic genes in HaCaT cells, including Bad, Bak, Diablo, Noxa, or Puma, and had only a minimal effect on the expression of caspases 3 and 9. Notably, these effects were not replicated in SiHa and HeLa tumor cell lines, where PTX did exert a significant proapoptotic effect.
Comparable results were reported in a study evaluating the combination of PTX and DOX in breast (MCF-7) and melanoma (MEL-Juso) tumor lines compared to HaCaT cells. In this case, individual PTX treatment showed virtually no effect on the viability of non-tumor cells, an effect that was not replicable in cancer cells.
Additionally, a similar protective profile has been documented in normal cardiac tissue from murine models treated with PTX and anthracyclines. In these studies, PTX administration did not induce apoptosis in cardiomyocytes and was able to attenuate anthracycline-induced apoptosis and reduce the expression of FasL and caspase-3 in this tissue.
The cytotoxic potential of PTX has also been evaluated in lymphocytes from healthy individuals. For instance, Loveena Rishi et al. isolated peripheral blood lymphocytes from healthy donors and treated them with various concentrations of PTX (0, 1.5, 3, 4.5, and 6 mg/mL) for 24 and 48 hours. In all cases, cell viability remained above 96%, demonstrating that even high PTX concentrations (up to 6 mg/mL) are not cytotoxic to normal lymphocytes (Discussion: p. 19, lines 625-626).
We acknowledge that the use of a single cell line represents a limitation. However, as demonstrated by other studies, Hs-445 has been increasingly recognized as a valuable model system for studying Hodgkin lymphoma. These studies have shown that Hs-445 effectively replicates the in vivo environment necessary for studying HL. Moreover, the HS-445 cell line has been used to explore the effects of various therapeutic agents, including chemotherapy and novel targeted therapies. These limitations have already been addressed in the main text of the manuscript (Discussion: p. 19, lines 633-638).
Regarding the use of primary HL samples, we consider that this approach has limitations inherent to their biological nature. It is well documented that the tumor microenvironment (TME) in HL is highly heterogeneous, with a complex cellular composition that varies among the different histological subtypes. This environment includes a minority population of malignant Hodgkin and Reed–Sternberg (HRS) cells (approximately 1% of total cells, with a maximum reported of 10%), surrounded by a diverse array of innate and adaptive immune cells. Consequently, direct assessment of PTX effects on tumor cells in primary samples may be confounded by the low frequency of HRS cells and the complexity of the surrounding

Reviewer 2 Report
Comments and Suggestions for Authors
In this manuscript the authors report repurposing of pentoxifylline (PTX) for treatment of Hodgkin lymphoma (HL). In particular, they investigated the potential of PTX for enhancing the anticancer activity of doxorubicin and bleomycin, which are used to treat HL. The manuscript is well written; however, the novelty of work is limited. Please see below for specific comments.
Reviewer Comments:
- Previously, PTX has shown antitumor activities in various tumor cell lines and based on its mechanism of action it is expected that PTX will also show anticancer effect in HL. Therefore, the novelty of work is limited. In particular the anticancer activity of PTX in HL was studied in only one cell line.
- Can the authors comment on the selectivity of PTX? Can PTX suppresses the expression of antiapoptotic genes that protect against cell death in noncancerous cells?
- In Figure 1 caption please indicate the method used to determine the IC50 for each drug (e.g., WST-1, etc.)
- Lines 105-109: can the authors elaborate more on the use of apoptosis assay employing flow cytometry to determine the IC50? Also, please make sure to include the methodology of this experiment in the manuscript. The IC50 value of BLM (i.e., 100 mU) was determined form this assay. Does Figure 1e was generated from this assay or the WST-1 assay?
- Figures 1a, 1b, 1f, 1g, 1h, 1i, 1j, and 1k do not add any value to the manuscript. Additionally, they are not cited in the manuscript. So, it is advisable to remove them from the manuscript or move them to the supplementary material if necessary.
- Line 118 and 122: there is an inconsistency. the authors mentioned that the triple therapy showed a synergetic effect on one occasion and an additive effect in another. Also, two values for the mean synergy score were reported (13.434 vs 9.7). The same comment applies for other combinations (lines 121-126). Please revise for clarity.
- Figures 3 and 4: it is not clear which bars are compared in terms of p-value.
- Lines 208-209: in fact, all tested combinations showed similar activation of caspase-8 and 9 and exceeded the bassline. can the authors explain the absence of synergetic effect of PTX on caspase enzyme activity in the triple combination?
- Line 100, 104 and 108: use IC50 instead of concentration.

Author Response
Comment 1: Previously, PTX has shown antitumor activities in various tumor cell lines and based on its mechanism of action it is expected that PTX will also show anticancer effect in HL. Therefore, the novelty of work is limited. In particular the anticancer activity of PTX in HL was studied in only one cell line.
Response 1: We sincerely appreciate your detailed and valuable feedback. Although the antitumor and proapoptotic effects of pentoxifylline (PTX) have been previously demonstrated in various tumor cell lines, to our knowledge, its specific effect in a Hodgkin lymphoma (HL) model had not been studied. Therefore, evaluating the direct impact of this molecule on HL is of particular interest, given the distinctive pathophysiology of this disease, which involves multiple mechanisms on which PTX has been shown to exert modulatory effects. In this context, PTX could represent a promising adjuvant strategy in the treatment of patients with HL.
In recent years, therapeutic advances in HL have primarily focused on immunotherapy, including monoclonal antibodies such as brentuximab vedotin and immune checkpoint inhibitors such as pembrolizumab. However, these agents are not yet approved as first-line
therapies. Current initial treatment regimens remain based on chemotherapy, with or without involved-field radiotherapy. Although ongoing clinical trials are evaluating the combination of immunotherapy with chemotherapy, such as brentuximab, in patients with advanced disease, its current use is primarily restricted to relapsed, refractory, or advanced-stage cases.
Moreover, immunotherapies face significant limitations related to high cost and limited availability, particularly in regions such as Latin America and Low- and Middle-Income Countries. Despite their promise, a considerable proportion of patients fail to achieve long-term remission, and cure rates remain suboptimal. Thus, it is essential to explore new therapeutic strategies that can be implemented from the early stages of the disease and that are also accessible and cost-effective. In this setting, PTX emerges as a viable alternative.
Among the most widely used chemotherapeutic agents for HL, anthracyclines and bleomycin stand out, both associated with severe adverse effects such as cardiotoxicity and pulmonary fibrosis. These toxicities can significantly increase morbidity and hinder treatment continuity. In this regard, the concomitant use of an agent like PTX, which not only enhances the cytotoxic effects of these drugs but may also mitigate their toxicity, represents a meaningful therapeutic innovation in the context of HL.
It is also noteworthy that, although PTX has been studied both alone and in combination with several chemotherapeutic agents, particularly anthracyclines, its combination with bleomycin (BLM) has, to our knowledge, not been previously evaluated in the cancer setting. Therefore, our study is the first to report this interaction, demonstrating that the PTX + BLM combination exerts a significant proapoptotic effect, comparable to that observed with the triple combination of PTX + DOX + BLM.
Comment 2: Can the authors comment on the selectivity of PTX? Can PTX suppresses the expression of antiapoptotic genes that protect against cell death in noncancerous cells?
Response 2: There is evidence demonstrating the protective effect of PTX on healthy tissues, as it does not induce cytotoxicity in non-tumor cells. For example, in several studies evaluating the effect of PTX in combination with cisplatin or adriamycin in cervical cancer cell lines, a non-tumor epithelial line (HaCaT) was also used. In these studies, various concentrations of PTX (1, 2, 4, 8, 16, and 20 mM) had no significant effect on cell viability compared to untreated controls, regardless of the administered dose. Moreover, no significant differences were observed in early or late apoptosis, caspase activation, or in the expression of proteins such as p-p38 and p-p65 following PTX treatment.
Similarly, when analyzing gene expression of various apoptosis-related markers, exclusive PTX treatment did not induce overexpression of proapoptotic genes in HaCaT cells, including Bad, Bak, Diablo, Noxa, or Puma, and had only a minimal effect on the expression of caspases 3 and 9. Notably, these effects were not replicated in SiHa and HeLa tumor cell lines, where PTX did exert a significant proapoptotic effect.
Comparable results were reported in a study evaluating the combination of PTX and DOX in breast (MCF-7) and melanoma (MEL-Juso) tumor lines compared to HaCaT cells. In this case, individual PTX treatment showed virtually no effect on the viability of non-tumor cells, an effect that was not replicable in cancer cells.
Additionally, a similar protective profile has been documented in normal cardiac tissue from murine models treated with PTX and anthracyclines. In these studies, PTX administration did
not induce apoptosis in cardiomyocytes and was able to attenuate anthracycline-induced apoptosis and reduce the expression of FasL and caspase-3 in this tissue.
The cytotoxic potential of PTX has also been evaluated in lymphocytes from healthy individuals. For instance, Loveena Rishi et al. isolated peripheral blood lymphocytes from healthy donors and treated them with various concentrations of PTX (0, 1.5, 3, 4.5, and 6 mg/mL) for 24 and 48 hours. In all cases, cell viability remained above 96%, demonstrating that even high PTX concentrations (up to 6 mg/mL) are not cytotoxic to normal lymphocytes.
Comments 3: In Figure 1 caption please indicate the method used to determine the IC50 for each drug (e.g., WST-1, etc.).
Response 3: Thank you for your comment. We have added the missing information to Figure 1, which now includes details of the assay used to determine the IC₅₀ (Results: p. 4, lines 158-159).
Comment 4: Lines 105-109: can the authors elaborate more on the use of apoptosis assay employing flow cytometry to determine the IC50? Also, please make sure to include the methodology of this experiment in the manuscript. The IC50 value of BLM (i.e., 100 mU) was determined form this assay. Does Figure 1e was generated from this assay or the WST-1 assay?
Response 4: We agree with your observation regarding the methodology for IC₅₀ determination. Accordingly, we have revised the corresponding section to clarify the procedure and prevent any confusion (Results: p. 3, lines 104-114).
Comment 5: Figures 1a, 1b, 1f, 1g, 1h, 1i, 1j, and 1k do not add any value to the manuscript. Additionally, they are not cited in the manuscript. So, it is advisable to remove them from the manuscript or move them to the supplementary material if necessary.
Response 5: We fully agree with your observation, so we have removed Figures 1a, 1b, 1g, 1h, 1i, 1j, and 1k, which do not add value to the manuscript, and have added them as supplementary material.
Comment 6: Line 118 and 122: there is an inconsistency. the authors mentioned that the triple therapy showed a synergetic effect on one occasion and an additive effect in another. Also, two values for the mean synergy score were reported (13.434 vs 9.7). The same comment applies for other combinations (lines 121-126). Please revise for clarity.
Response 6: We sincerely appreciate your feedback on the presentation of these results. We acknowledge that the original wording may have led to perceived inconsistency; therefore, we have rephrased the text to improve clarity (Results: pp. 3-4, lines 128-153).
Comment 7: Figures 3 and 4: it is not clear which bars are compared in terms of p-value
Response 7: Thank you for highlighting this point. We agree that it is unclear which bars are being compared in terms of p-value in Figures 3 and 4. We have therefore modified the figures to provide a more precise and more accurate representation of the data (Results: pp. 6, and 7, Figures 3, and 4).
Comment 8: Lines 208-209: in fact, all tested combinations showed similar activation of caspase-8 and 9 and exceeded the bassline. can the authors explain the absence of synergetic effect of PTX on caspase enzyme activity in the triple combination?
Response 8: We sincerely appreciate your observation. The synergistic effect observed with the triple combination of PTX (8 mM), DOX (1 µM), and BLM (100 mU) was determined from
the cell viability assay, which globally reflects the pharmacological interaction between the compounds in terms of their ability to induce cell death. It is important to note that specific analyses were not performed to evaluate synergy or additivity in each of the complementary assays (e.g., caspase activity, mitochondrial membrane potential, cell proliferation, among others).
In this sense, although the activity of caspases 8 and 9 was similar in all the combinations evaluated, this does not necessarily contradict the synergistic effect observed on cell viability. Cell death is a multifactorial process that does not depend exclusively on the activation of these caspases, but can result from the integration of various apoptotic and non-apoptotic pathways.
Indeed, gene expression analyses showed that the triple combination induced the highest upregulation (either positive or negative) of genes related to apoptosis, cell cycle, and signaling pathways, such as BCL-2, P21, P16, and P65, suggesting that multiple mechanisms could be involved in the synergistic response observed with the triple combination. Therefore, the absence of differential activation of caspases compared to the other combinations does not exclude the presence of a synergistic effect; instead, it emphasizes the importance of considering cell death as the result of a set of pathways and mechanisms that extend beyond their individual elements.
Comment 9: Line 100, 104 and 108: use IC50 instead of concentration.
Response 9: We agree with your suggestion to use the term IC₅₀ instead of "concentration". The formatting has been corrected, and the term IC₅₀ has been applied in all suggested lines (Results: p. 3, lines 100, 103, and 113).
